# Spatiotemporal variation of aerosol and potential long-range transport impact over Tibetan Plateau, China

Jun Zhu[1,2,3], Xiangao Xia[2,4], Huizheng Che[3], Jun Wang[5], Zhiyuan Cong[6], Tianliang Zhao[1], Shichang Kang[7,9], Xuelei Zhang[8], Xingna Yu[1], Yanlin Zhang[1]

[1] Collaborative Innovation Center on Forecast and Evaluation of Meteorological Disasters, Key Laboratory for Aerosol-Cloud-Precipitation of China Meteorological Administration, Nanjing University of Information Science and Technology, Nanjing 210044, China;

[2] LAGEO, Institute of Atmospheric Physics, Chinese Academy of Sciences, Beijing 100029, China;

[3] State Key Laboratory of Severe Weather (LASW) and Key Laboratory of Atmospheric Chemistry (LAC), Chinese Academy of Meteorological Sciences, CMA, Beijing, 100081, China;

[4] University of Chinese Academy of Sciences, Beijing, 100049, China;

[5] Center of Global and Regional Environmental Research and Department of Chemical and Biochemical Engineering, University of Iowa, Iowa City, Iowa, USA;

[6] Key Laboratory of Tibetan Environment Changes and Land Surface Processes, Institute of Tibetan Plateau Research, Chinese Academy of Sciences, Beijing 100101, China;

[7] State Key Laboratory of Cryospheric Science, Northwest Institute of Eco-Environment and Resources, Chinese Academy of Sciences, Lanzhou 730000, China;

[8] Northeast Institute of Geography and Agroecology, Chinese Academy of Sciences, Changchun 130102, China;

[9] CAS Center for Excellence in Tibetan Plateau Earth Sciences, China.

Corresponding author: Jun Zhu (junzhu@nuist.edu.cn) & Xiangao Xia (xxa@mail.iap.ac.cn)

**Abstract:**

The long-term temporal-spatial variations in the aerosol optical properties over the Tibetan Plateau (TP) and the potential long-range transport from surrounding areas to TP were analysed in this work, by using multiple years of sunphotometer measurements (CE318) at five stations in the TP, satellite aerosol products from the Moderate Resolution Imaging Spectroradiometer (MODIS) and Cloud-Aerosol Lidar with Orthogonal Polarization (CALIOP), back-trajectory analysis from the Hybrid Single-Particle Lagrangian Integrated Trajectory (HYSPLIT) and model simulations from the Goddard Earth Observing System (GEOS)-Chem chemistry transport model. The results from the ground-based observations showed that the annual aerosol optical depth (AOD) at 440 nm at most TP sites increased in recent decades with trends of $0.001\pm0.003$/year at Lhasa, $0.013\pm0.003$/year at Mt_WLG, $0.002\pm0.002$/year at NAM_CO, and $0.000\pm0.002$/year at QOMS_CAS. The increasing trend was also found for the aerosol Extinction Ångstrom exponent (EAE) at most sites with the exception of the Mt_WLG site. Spatially, the AOD at 550 nm observed from MODIS showed negative trends at the northwest edge close to the Taklimakan Desert and to the east of the Qaidam Basin and slightly positive trends in most of the other areas of the TP. Different aerosol types and sources contributed to a polluted day (with CE318 AOD at 440 nm > 0.4) at the five sites on the TP: dust was dominant aerosol type in Lhasa, Mt_WLG and Muztagh with sources in the Taklimakan Desert but fine aerosol pollution was dominant at NAM_CO and QOMS_CAS with transport from

South Asia. A case of aerosol pollution at Lhasa, NAM_CO and QOMS_CAS during 28 April – 3 May 2016 revealed that the smoke aerosols from South Asia were lifted up to 10 km and transported to the TP, while the dust from the Taklimakan Desert could climb the north slope of the TP and then be transported to the central TP. The long-range transport of aerosol thereby seriously impacted the aerosol loading over the TP.

**Keywords:** Aerosol optical depth, Tibetan Plateau, aerosol pollution, long-range transport

**1. Introduction**

The heavy haze that has occurred in recent years in China has been largely attributed to the atmospheric aerosols (Zhang et al., 2015). In addition, atmospheric aerosols can affect the climate through the interactions between aerosol-radiation and between aerosol-cloud (Takemura et al., 2005; Li et al., 2017; Huang et al., 2006a; Huang et al., 2006b; Liu et al., 2014; Liu et al., 2011), while the clouds and its precipitation are also connected to large scale atmospheric circulations (Yang et al., 2010; Yang et al., 2017a). However, there is still a high level of uncertainty about the impact of aerosols on the climate, which is mostly due to the high spatiotemporal variability of aerosols (IPCC, 2013). Therefore, studying the physical and chemical properties of aerosols over different regions is essential.

The Tibetan Plateau (TP), is the largest elevated plateau in East Asia and considered as one of the most pristine terrestrial regions, along with the Arctic and Antarctic. However, in the past two decades, the TP has been surrounded by an unprecedented growth of emissions of Asian air pollutants from various sources. Consequently, some studies have demonstrated that the aerosols transported from its around areas (South Asia and Taklimakan Desert) have polluted the TP (Huang et al., 2007; Xia et al., 2011; Kopacz et al., 2011; Lu et al., 2012; Liu et al., 2015; Zhu et al., 2018; Jia et al., 2019; Jia et al., 2015). The increase in aerosols over the TP may have an important impact on the regional or global climate. Lau et al. (2006) has suggested that increased absorbing aerosols (dust and black carbon) over the TP may create a positive tropospheric temperature anomaly over the TP and adjacent regions to the south, causing the advance and enhancement of the Indian summer monsoon. Besides the impact of aerosols over the TP on the radiation budget, temperature and Indian summer monsoon, Liu et al. (2019a) reported a potential relationship may exist between the aerosol index and ice cloud properties over the TP, in which the aerosols have a more dominant influence than meteorological conditions on ice cloud properties (except for the nocturnal ice cloud droplet radius and ice water path during the daytime). Furthermore, Liu et al. (2019b) found the effect of the dust aerosols over the TP on the development of convective clouds and then movement of the some developed convective clouds could induce significant precipitation over the Yangtze River basin and North China. Attempts have been made to reveal the linkages between climate change (such as changes to glaciers and monsoons) and the air pollutants around the TP (mainly absorbing carbonaceous materials) (Qian et al., 2011; Wang et al., 2016; Lee et al., 2013; Jia et al., 2018). However, the quantitative effect of the TP aerosol on climate variability remains largely unknown, and it is very essential to fully understand the aerosol characteristics over the TP.

A large amount of attention has been paid to aerosol characteristics over the TP (Wan et al., 2015; Tobo et al., 2007; Zhao et al., 2013; Liu et al., 2008; Du et al., 2015). Although the seasonal variations in aerosol properties over the TP have been analysed based on ground-based observations or satellite products (Shen et al., 2015; Xia et al., 2008), analysis is needed of the long-term trends in the variation of aerosols over the TP to provide predictions and guidelines for environment policies. In past studies, spring or summer have often been studied due to the important impacts of dust and carbonaceous aerosols (Huang et al., 2007; Cong et al., 2007; Lee et al., 2013). However, most studies of the aerosol properties based on ground-based measurements have been conducted at a single site over the TP, such as NAM_CO (Cong et al., 2009), Mt_Yulong (Zhang et al., 2012), and Mt_WLG (Che et al., 2011). Past studies have mostly focused on single stations or short-term

variations due to the difficulty of taking a sufficient number of ground-based observations in challenging weather conditions over the remote plateau.

Ground-based measurements can offer more accurate data on aerosol properties, while large-scale spatial observations of aerosol optical and physical properties require satellite remote-sensing methods (Li et al., 2015; Li et al., 2018; Xing et al., 2017). Thus, the long-term detection of aerosols from both ground and satellite platforms is absolutely necessary for improving our understanding of the climate effects of aerosol over the TP region. Consequently, based on multiple years of observations from five ground-based sunphotometers at the TP and the MODIS aerosol optical depth product over the TP region, our work here is focused on the long-term spatiotemporal variations in the aerosol optical properties over the TP and the aerosol properties and sources during the high aerosol loading over the TP. In addition, we also combined the observation and models to study the aerosol transport process over the TP, thereby helping to reduce the uncertainties in estimating aerosol radiative forcing and aerosol sources.

In this paper, section 2 describes the observation sites, data and methods. The results of the analysis of the spatiotemporal variations in aerosol properties over the TP are shown in section 3. The analysis of aerosol high loading and an aerosol transport case are presented in section 4 and 5, respectively. The conclusions are presented in section 6.

## 2. Site, data and methodology
### 2.1 Sites
In this study, five sites in the TP equipped with sun and sky scanning radiometers (CE318) were used (Figure 1). Table 1 shows the station locations and descriptions. Lhasa station is the only urban site that suffers from local anthropogenic emissions. For the other four sites, local anthropogenic emissions are extremely rare due to the low number of human inhabitants. However, Mt_WLG is in the northeast of the TP, where it is situated on the dust transport path from the largest desert in China (the Taklimakan Desert). The Muztagh_Ata site is located in the northwest corner of the TP and next to the Central Asian Deserty and the Taklimakan Desert. NAM_CO is located on the central Tibetan Plateau, 220 km away from Lhasa. QOMS_CAS is located at the northern slope of Mt. Qomolangma on the border of Tibet and Nepal. Therefore, these five sites are representative of the spatial features of the TP.

### 2.2 Data
2.2.1 CE318 aerosol optical properties
The column-integrated aerosol properties over the five TP sites are derived from CE318 measurements. Table 1 shows the observation period. The CE318 instrument measures direct solar spectral radiation and the angular distribution of sky radiance. These spectral radiances can be used to retrieve aerosol optical parameters (such as aerosol optical depth (AOD)) based on Beer's Law and aerosol microphysical properties (such as volume size distribution) and the radiative forcing through radiation transfer theory (Dubovik and King, 2000; Dubovik et al., 2006). The instruments were periodically calibrated using the Langley method at AERONET global calibration sites (the Izaña, Spain or the Mauna Loa, USA) or using the inter-comparison calibration method at the Beijing-CAMS site (Che et al., 2015). The cloud-screened and quality-controlled data of AOD,

Extinction Ångstrom exponent (EAE), and aerosol volume size distribution (dV(r)/dlnr) are used in this work (Giles et al., 2019). Eck et al. (1999) showed that the uncertainty of the AOD was approximately 0.01 to 0.02. The EAE was calculated from the AOD at 440 and 870 nm. The errors of retrieval for dV(r)/dlnr were less than 10% in the maxima of the dV(r)/dlnr and may increase up to 35% for the minimum values of dV(r)/dlnr within the radius range between 0.1 μm and 7 μm; for the edges of the retrieval size, the errors increased apparently, but did not significantly affect the derivation of the main feature of dV(r)/dlnr (Dubovik et al., 2002).

### 2.2.2 The MODIS AOD product

The Moderate Resolution Imaging Spectroradiometer (MODIS) instrument is a multi-spectral sensor with a wide spectral range from 0.4 to 14.4 μm in 36 wavelength bands, onboard the Terra (morning descending direction) and Aqua (afternoon ascending direction) satellites in polar orbit. It's broad swath of 2330 km permits retrieval aerosol products to cover the global word within 1-2 days. In this study, both Terra and Aqua MODIS Collection 6 Deep-Blue (DB) and Dark-Target (DT) combined AOD at 550 nm product with 10 km spatial resolution (Levy et al., 2013) from 2006 to 2017 were used. The combined MODIS DT and DB AOD at 550 nm (MODIS_AOD) merges the products from the two algorithms based on the normalized difference vegetation index (NDVI) statistics as follows: 1) the DT AOD data are used for NDVI > 0.3; 2) the DB AOD data are used for NDVI < 0.2; and 3) the mean of both the algorithms or AOD data with a high quality flag are used for $0.2 \leq$ NDVI $\leq 0.3$. The MODIS_AOD had been validated in the global or regional areas (Bilal et al., 2018; Ma et al., 2016; Sayer et al., 2014). The root-mean-square error (RMSE) of MODIS_AOD was approximately 0.13, and the percentage of MODIS_AOD data within the expected error was more than 71% at the Kunming site, which is near the TP (Zhu et al., 2016).

### 2.2.3 The CALIOP profile data

The Cloud-Aerosol Lidar with Orthogonal Polarization (CALIOP), the primary instrument on board the CALIPSO satellite, is a near-nadir-viewing two-wavelength (532 nm and 1064 nm) polarization-sensitive lidar that performs global vertical profile measurements of aerosols and clouds (Winker et al., 2010). It provides three primary calibrated and geolocated profile products: total attenuated backscatter at 532 nm and 1064 nm and the perpendicular polarization component at 532 nm. The CALIOP products (version 4.10) used in this study include the attenuated backscattering coefficient profiles from Level 1B and the vertical feature mask data products of aerosol subtype from Level 2 products under 15 km altitude, which were downloaded from the Langley Atmospheric Science Data Center (ASDC). Kumar et al. (2018) had showed that the AOD from CALIOP version 4.10 agreed with the ground-based CE318 observation at a site in the central Himalayas with a correlation > 0.9 and ~ 87 % matchup data were within the expected error limits.

### 2.3 Methodology

The ground-based CE318 observations and MODIS AOD products were analysed to show the spatiotemporal variations in aerosol properties in the TP.

The CE318 observed AOD at 440 nm with values larger than 0.4 at each site was specially analysed to study the aerosol properties of the high aerosol loading over the TP. The value of 0.4 was selected because the mean annual values of AOD observed by CE318 instruments at the TP

sites were less than ~0.1 in the past studies (Xia et al., 2016; Cong et al., 2009), and this value is normally regarded as the high aerosol loading (Eck et al., 2010; Giles et al., 2012). Back trajectories were used for the aerosol source analysis in the TP. The back trajectories on the high aerosol loading days were calculated by using the Hybrid Single-Particle Lagrangian Integrated Trajectory (HYSPLIT) model which was driven by the one degree horizontal resolution archived meteorological fields (Draxler and Hess, 1998). 72-hour back trajectories ending at the five sites at 10 m above ground level at 12 UTC on the days with high aerosol loading (CE318 AOD at 440 nm >0.4) were used to identify the air mass sources.

A case of long-range aerosol transport to the TP was selected based on the ground CE318 observations over Lhasa, NAM_CO and QOMS_CAS. The HYSPLIT back trajectories and the MODIS and CALIOP products were used to show the potential aerosol sources, spatial aerosol loading and the vertical features of the aerosol over the TP during the case period. In addition, the Goddard Earth Observing System (GEOS)-Chem chemistry transport model was used to simulate the AOD and its components (dust and carbon aerosols) during the case period, which may reflect the change in aerosol type during the case period.

The GEOS-Chem chemical transport model (version 11-01) coupled with the online radiative transfer calculations (RRTMG) at $0.5° \times 0.667°$ horizontal resolution over East Asia domain (Bey et al., 2001; Wang et al., 2004) was used. The model was driving by the Global Modeling and Assimilation Office (GMAO) MERRA-2 meteorology with the temporal resolution of 3 hours for meteorological parameters and 1 hour for surface fields. The simulation type of full chemistry in the troposphere was selected. The implementation of RRTMG in GEOS-Chem was described in Heald et al. (2014). The AOD was calculated according to Martin et al. (2003). The default global anthropogenic emissions were overwritten over East Asia by the MIX inventory from Li et al. (2014). The Global Fire Emission Database (GFED) (van der Werf et al., 2010) has been used to specify emissions from fire. More details on the model and the other emissions data used and the evaluation of AOD in the east and south of the TP were shown in Zhu et al. (2017).

In this study, the AOD from the CE318, MODIS, and GEOS-Chem model were used. For convenience, CE318_AOD, MODIS_AOD, and Model_AOD stand for the AOD observed by CE318, MODIS, and the AOD simulated by the GEOS-Chem model, respectively. For CE318_AOD, the 440 nm wavelength is often studied, while MODIS_AOD and Model_AOD generally use the data at 550 nm wavelength. Thus, unless otherwise specified, CE318_AOD, MODIS_AOD, and Model_AOD hereinafter represent the ones at 440 nm, 550 nm, and 550 nm, respectively.

**3. Temporal-spatial variations in aerosol properties**
**3.1 Aerosol properties observed by the CE318 instruments**
The monthly, seasonal, and annual variations in aerosol properties observed from the CE318 instruments at the five TP sites were analyzed.

The monthly variations in CE318_AOD and EAE at the five sites over the TP are shown in Figure 2. The monthly mean CE318_AOD was highest in April at the Lhasa (0.19), NAM_CO (0.09)

and QOMS_CAS (0.10) sites, while the value at Mt_WLG was highest in June (0.20). The monthly mean CE318_AOD rapidly increased from January to April and then slightly decreased until December at the Lhasa, NAM_CO and QOMS_CAS sites. However, the monthly mean CE318_AOD at Mt_WLG was nearly symmetrical form from January to December. The monthly variation in EAE was different from the AOD. The highest monthly mean values of EAE occurred in September at Lhasa (1.15), October at Mt_WLG (1.15) and in January at the NAM_CO (0.93) and QOMS_CAS (0.17) sites. The EAE at QOMS_CAS also showed a high value of 0.17 in April, which may be caused by the smoke aerosol transported from South Asia during this period. The monthly mean EAE first decreased from January to March and then increased until September at Lhasa. The monthly mean EAE values at NAM_CO also decreased from January to March, but did not increase apparently in the following months. The EAE at Mt_WLG decreased from January to May and then increased obviously from May to October. The Lhasa, NAM_CO, and QOMS_CAS sites are near and located in the south of the TP. Thus, the variations in the aerosol properties at these three sites were similar. The Mt_WLG site is located in the northeast of the TP, which is different from the southern sites. The Muztagh_Alt is in the northwest of the TP and is the closest site to the Taklimakan Desert, which causes the high AOD there (a few observed data may be another reason). Looking at the monthly CE318_AOD and EAE values together, high CE318_AOD was often accompanied by low EAE at Lhasa, Mt_WLG and NAM_CO, indicating that these sites suffered from coarse aerosols such as dust (Huang et al., 2007; Liu et al., 2015; Zhang et al., 2001). However, the QOMS_CAS site showed high CE318_AOD and high EAE in April, which may be related to smoke aerosols transported from South Asia.

Table 2 shows the seasonal statistics of CE318_AOD and EAE. A distinct seasonal variation in CE318_AOD and EAE can be found over the TP sites. The CE318_AOD mean values in fall (SON) and winter (DJF) were lower at all sites except Muztagh. Muztagh_Ata showed high CE318_AOD in both observed seasons. Except for that in Muztagh, the maximum seasonal CE318_AOD was observed in spring (MAM) (Lhasa, NAM_CO, and QOMS_CAS) or in summer (JJA) (Mt_WLG). The minimum seasonal EAE occurred in spring (Lhasa and Mt_WLG) or summer (NAM_CO and QOMS_CAS), while the maximum EAE values were mostly observed in fall (Lhasa and Mt_WLG) and winter (NAM_CO and QOMS_CAS). These indicated frequent dust events over the TP in the spring and summer period. Mt_WLG is situated on the dust transport path from the Taklimakan Desert, which causes the high CE318_AOD observed in spring and summer at this site.

The seasonal size distributions of the five sites in Figure 3 also demonstrated that coarse mode aerosol was dominant at the five TP sites in almost all seasons, which was different from those in the eastern pollution regions of China, such as Yangtze River Delta, where fine mode aerosol was dominant (Zhuang et al., 2018). This size distribution explained the relatively low annual averages of EAE at the five sites (all annual EAE in Figure 2 are less than 1.0), compared to the those at the inland urban and suburban sites in China (Xin et al., 2007), such as Beijing (1.19) (Fan et al., 2006), Nanjing (1.20) (Zhuang et al., 2018; Zhuang et al., 2017), Kunming (1.25) (Zhu et al., 2016), and Chengdu (1.09) (Che et al., 2015). What's more, spring was the season with a high volume concentration of coarse mode aerosol. Among the five sites, the southernmost sites, QOMS_CAS, showed the highest annual mean EAE and the size distribution was distinctly bimodal, especially in spring. This was also because of the frequent biomass burning activity in India and Nepal, which

can transport the fine aerosol to the QOMS_CAS site.

The annual averages of CE318_AOD (shown in Figure 2) were 0.05-0.14 over TP sites. These average values were lower than those in other regional background sites, such as Longfengshan (0.35) in Northeast China (Wang et al., 2010), Xinglong (0.28) in North China Plain (Zhu et al., 2014), Lin'an (0.89) in East China (Pan et al., 2010) and Dinghushan (0.91) in South China (Chen et al., 2014). The low aerosol loading over the five TP sites indicates excellent air quality over the TP region.

However, the aerosol loading at the TP sites presented interannual changes. The annual variations in CE318_AOD and EAE over the TP at the four sites, i.e. Lhasa, Mt_WLG, NAM_CO, and QOMS_CAS are shown in Figure 4. The data for the CE318 observations at Muztagh_Ata site are only available for 2010; thus, the annual variation at this site is not shown here. The annual CE318_AOD showed increasing trends of $0.001 \pm 0.003$/year at Lhasa, $0.013 \pm 0.003$/year at Mt_WLG, and $0.002\pm0.002$/year at NAM_CO during the CE318 observation period. The Mt_WLG site showed the most obvious increase in CE318_AOD during 2009-2013. These results indicated an increase in aerosol loading at the three sites. The long-term annual variation of CE318_AOD at QOMS_CAS was very small ($0.000\pm0.002$/year), but there were still short-term annual variations (the values decreased from 2010 to 2013 and increased from 2013 to 2016). The annual trends of EAEs were more evident than the CE318_AOD at these four site. Most sites showed an increasing tendency in the average annual EAE except for Mt_WLG site, which showed a large decreasing trend of $-0.318\pm0.081$/year. This showed that the size of aerosol at the Mt_WLG site increased, while the size of the aerosol decreased in the other three sites. Looking at the CE318_AOD and EAE values together, the positive trend of CE318_AOD and the positive trend of EAE in the long term variation at most sites over the TP indicated the addition of fine mode aerosol which may be related to the anthropogenic impact or long-distance transport of polluted dust to the TP. However, in the short term, the increase in the average annual CE318_AOD was often associated with the decrease in EAE over the TP, which suggested the addition of coarse mode aerosol during the CE318 observation period.

**3.2 Aerosol properties from MODIS**

Ground-based observations can offer accurate aerosol optical properties at point locations but lack spatial coverage. The MODIS aerosol product can provide the spatial variation in AOD over the TP. Thus, we evaluated the MODIS_AOD using the ground-based observation CE318_AOD at 550 nm over the TP sites. The CE318_AOD at 550 nm was interpolated from 440 nm, 675 nm, 870 nm and 1020 nm by using an established fitting method from Ångström (1929). The matchup method was that the CE318 data within 1 hour of the MODIS overpass were compared with the MODIS data within a 25 km radius of the ground-based site. The minimum requirement for a matchup was at least 3 pixels from MODIS.

Figure 5 shows the results of MODIS_AOD compared to the collocated ground CE318 observations over the TP. There were 996 instantaneous matchups of Terra and Aqua MODIS during the CE318 instrument measurement period at the five TP sites. The MODIS_AOD overestimated the AOD at 550 nm with a positive mean bias of 0.02 and a RMSE of 0.11. The RMSE value was

lower than that of the North China Plain sites (~0.25) (Bilal et al., 2019). The slope and intercept of the best-fit equation between the MODIS_AOD and CE318_AOD at 550 nm were 0.46 and 0.06, respectively, with a correlation coefficient (R) of 0.54. There were 67.8% of the compared AODs within the expected error envelope of 0.05+0.15AOD (%EE). The R value was lower than that in the global assessment statistics, while the %EE was higher than that in the global evaluation (Bilal and Qiu, 2018). Overall, the results suggest that the MODIS_AOD product can be used to study the aerosol spatial variation over the TP region.

The spatial distribution of the annual MODIS_AOD is shown in Figure 6. The MODIS_AOD agreed with the CE318_AOD at 550 nm at the five TP sites. The northwest area around the Taklimakan Desert and the northern part of the TP on the transport path of the Taklimakan Desert dust showed high MODIS_AOD (>0.25) in recent decades. In addition, the southern edge performed slightly high MODIS_AOD (0.2-0.25) influenced by the aerosol transport from South Asia. There were some small areas with high MODIS_AOD (~0.2) in the centre of the TP, and the southeast region showed low MODIS_AOD (~0.1), which may be attributed to the aerosol transport and surface features such as vegetation cover, since there are few inhabitants. The seasonal departure of MODIS_AOD (Figure 7) showed that high positive MODIS_AOD departure often appeared in spring, especially for the northwest edge, northern area and southern edge of TP, which may be a result of aerosol transport from the frequent dust events in the Taklimakan Desert and the fire activities in South Asia in spring.

A linear regression analysis of the trends in annual MODIS_AOD over the TP from 2006 to 2017 was conducted using the least squares method. The spatial distribution of the annual trends in MODIS_AOD during 2006-2017 is illustrated in Figure 8. There were no statistically significant trends in most areas during 2006-2017. The MODIS_AOD showed negative trends in the northwest edge close to the Taklimakan Desert and to the east of the Qaidam Basin and slightly positive trends in most of the other areas. The areas where MODIS_AOD decreased were mainly located near the desert or on the transport path of the desert dust. This descending trend may be related to the significant reduction in dust emissions caused by the decline in wind speed in recent years (Yang et al., 2017b). The positive trend in other most areas may be due to the rapid increase in human activities, such as the expansion of tourism to the TP and biomass burning in South Asia.

The seasonal trends in MODIS_AOD at 550 nm over the TP during 2006-2017 are presented in Figure 9. The spring showed the most obvious decline in MODIS_AOD (~ 0.02/year) at the northern edges and northeast part of the TP during 2006-2017, which also suggested that the reduction in dust impact from the Taklimakan Desert like the trend in the annual MODIS_AOD (seen in Figure 8). In summer, the positive trend in MODIS_AOD over the TP was relatively apparent, and most sporadic higher positive values of ~0.01 occurred in the central and southern part of the TP. Summer is the tourist season in the TP and tourism has developed in past decades, which may be one of the reasons for the higher positive trend in summer in the TP. The positive trends in autumn and winter were relatively lower than those in summer, and the most positive trends were located at the northern TP. The reason for this phenomenon needs to be explored.

**4. Aerosol properties and potential sources during high aerosol loading**

The annual mean AOD in the TP was normally low due to the few human inhabitants and high altitude. However, some high CE318_AOD values larger than 0.4, which is normally regarded as high aerosol loading (Eck et al., 2010; Giles et al., 2012), were observed at the five sites in the TP by CE318. Thus, the CE318_AOD larger than 0.4 over TP can be considered as the aerosol pollution. The frequencies of high aerosol loading (CE318_AOD > 0.4) during the CE318 measurements were 1.57%, 1.79%, 0.21%, 0.42% and 0.11% at the Lhasa, Mt_WLG, Muztagh_Ata, NAM_CO, and QOMS_CAS sites, respectively. The aerosol properties and sources during high aerosol loading in the TP need to be studied.

Figure 10 shows the CE318_AOD with values larger than 0.4 versus EAE observed by CE318 at the five sites in the TP. Except for the Lhasa and Mt_WLG sites, almost all values of CE318_AOD were less than 1.0, which reflected the relatively clear environment over the TP. The EAE showed two centres at ~0.1 and ~1.5. The low EAE (~0.1) centre was related to dust events, which can cause higher concentrations of coarse particles in the atmosphere. Besides, most values of the low EAE (<0.5) area were less than 0.2 (only a few EAEs between 0.2-0.5 were observed at Lhasa and Mt_WLG), indicating that the pure dust type aerosols were more common than the polluted dust type aerosols in the TP according to Eck et al. (2010). The high EAE centre at ~1.5 indicateed mainly small sub-micron radius particles, which can be attributed to anthropologic emissions. The values of EAE >1.0 at the NAM_CO and QOMS_CAS sites were generally higher than those at the Lhasa and Mt_WLG sites. According to past studies, the EAE of biomass burning aerosol is generally higher than the urban/industry aerosol (Giles et al., 2012; Eck et al., 2010), which may cause the higher EAE at NAM_CO and QOMS_CAS (more biomass burning aerosol) than at Lhasa and Mt_WLG (more urban/industry aerosol) for the high EAE area. On the other hand, the values in the middle range of 0.5-1.0 were rare, indicating a smaller mix of natural and human sources. The percentage of EAE bins to the number of CE318_AOD>0.4 was distinct (Table 3). The percentage of EAE <0.5 was high than that of EAE>1.0 at Lhasa, Mt_WLG and Muztagh_Ata, indicating more nature dust pollution than the anthropogenic pollution at these three sites. However, a greater number of high EAE values (>1.0) were observed than EAE<0.5 values at the NAM_CO and QOMS_CAS sites, suggesting that anthropogenic pollution was more than natural dust pollution at these two sites.

Figure 11 shows the aerosol size distribution binned by CE318_AOD at the five sites in the TP. The volume concentration of coarse mode particles increased more apparently than fine mode at Lhasa, Mt_WLG and Muztagh sites when the values of CE318_AOD increased. However, the size distribution at NAM_CO and QOMS_CAS showed the dominant increase of fine mode aerosol. These indicate the different aerosol type pollution in these five sites: dust dominant in Lhasa, Mt_WLG and Muztagh_Ata and fine mode aerosol (mainly biomass burning aerosol) pollution dominant at NAM_CO and QOMS_CAS.

The dominant aerosol pollution type showed obvious distinctions among the five sites on the TP, then what was the distinct aerosol pollution source at each site? We used the HYSPLIT back-trajectory model and the MODIS_AOD on the day with aerosol pollution (CE318_AOD >0.4) to show the aerosol source for the pollution day at each site. Figure 12 shows the 72-hour back-trajectories ended at the five sites (10 m above ground level) in the TP overlaid with the mean

MODIS_AOD on the aerosol pollution day which was observed by the ground-based CE318 (CE318_AOD > 0.4). The CE318 instruments observed 78, 20, 2, 15, and 14 days at Lhasa, Mt_WLG, Muztagh_Ata, NAM_CO and QOMS_CAS, respectively, with instantaneous CE318_AOD > 0.4. The aerosol pollution days at Lhasa, Mt_WLG, and Muztagh_Ata observed by CE318 often had low EAE (black lines). The airflows ended at the Lhasa site on the polluted days were mainly from the northwest and southwest. The MODIS_AOD around Lhasa in the area of the back-trajectories with CE318 EAE <0.5 passing did not show significantly high values, especially in the Taklimakan Desert, which indicated that the dust pollution at Lhasa was mainly from local or surrounding dust events rather than transport from the Taklimakan Desert. The Mt_WLG showed that the air mass on the pollution days come from the west and east and the path of back trajectories had high MODIS_AOD. The high values of MODIS_AOD showed two transport paths of dust aerosol to Mt_WLG: one was through the Qaidam Basin and the other was through the northeast edge of the TP. The two polluted days observed by CE318 at the Muztagh_Ata showed the easterly airflows originating from the Taklimakan Desert. The direction of the back-trajectories of EAE<0.5 that ended at NAM_CO was similar to Lhasa, while the southerly air flows with high EAE (red trajectories) originated from Nepal, where frequent biomass burning happened and caused the high MODIS_AOD values. The trajectories ended at QOMS_CAS and the high MODIS_AOD of the path revealed the transport of finer aerosol from South Asia to this site.

## 5. Case study of long-range transport to the TP

The long-range transport of aerosol can cause the aerosol pollution and affect the long-term variation in aerosol over the TP. In addition, the dominant aerosol type may change at the TP sites during a case of aerosol transport. Thus, a specific case of aerosol pollution during 27 April−3 May 2016 was analysed further. This case was selected based on the observations from the CE318 instruments. During 28 April−1 May, the instantaneous CE318_AOD at Lhasa, NAM_CO, QOMS_CAS sites showed the values larger than 0.4, which reached up to more than 3 times the mean values of CE318_AOD of each site (0.11 at Lhasa, 0.05 at NAM_CO and QOMS_CAS). This was indicative of aerosol pollution at the three sites. Then, how about the aerosol properties of this period and where did the polluted aerosol originate?

Figure 13 shows the daily CE318_AOD and EAE during 27 April−3 May at the three sites. The mean values of CE318_AOD were 0.45, 0.38, and 0.23 at Lhasa, NAM_CO and QOMS_CAS, respectively. These even reached 4 times the annual mean CE318_AOD at each site. The mean EAEs were 0.98, 1.22, and 1.44 at Lhasa, NAM_CO and QOMS_CAS, respectively, which were higher than the annual averages and suggested the arrival of fine aerosols. There were CE318_AOD peaks at the three sites during 27 April−3 May. Lhasa showed an increase in CE318_AOD from 0.30 on 27 April to 0.51 on 28 April and maintained high CE318_AOD to a value of 0.54 on 1 May, after which it decreased to 0.34 on 2 May. NAM_CO also showed an increase of CE318_AOD during the first two days of the period, but decreased after 29 April. QOMS_CAM showed a slight increase in CE318_AOD from 27 April to 30 April, which was later than those of the other two sites. Combining the EAE on these days, fine mode aerosol was brought to Lhasa and NAM_CO during 27−29 April, and then coarse aerosol began to occur on 30 April, and even became the dominant aerosol in the following several days. The fine aerosol at the QOMS_CAM site was maintained for an additional day after those at the two sites, and then the coarse aerosol increased.

The GEOS-Chem model simulation also supported the above results. Figure 14 shows the comparison between the Model_AOD (0.5° × 0.667°) and CE318_AOD at 550 nm and the ratios of the model simulated aerosol types (dust, both organic carbon (OC) and black carbon (BC) aerosol) to the total Model_AOD during this case period at the three sites. The evaluation results showed that the model underestimated the daily AOD at the three sites during this period, with negative mean biases from -0.28 to -0.08. However, the Model_AOD was relatively high correlated with the CE318_AOD at 550 nm, with the R values of 0.61 at Lhasa, 0.89 at NAM_CO and 0.86 at QOMS_CAS. These R values were higher than the model evaluation in South China and the Indo-China Plain (~0.5) (Zhu et al., 2017). Thus, the variation trend from Model_AOD agreed well with that measured by the CE318 instruments during these days. During the first 4 days of the case period (27 April to 30 April), the ratios of different aerosols to the total Model_AOD showed that the sum of OC and BC aerosols were higher than those of dust aerosol at all three sites. Besides, the sums of OC and BC at Lhasa and QOMS_CAS were higher than that of NAM_CO. These indicated that the smoke aerosol affected the three sites more severely than dust during the first 4 days and Lhasa and QOMS_CAS sites were nearer to smoke sources than NAM_CO. After 30 April, the sum of BC and OC decreased while dust increased, and the increase of dust at the three sites was NAM_CO > Lhasa > QOMS_CAS. Therefore, the major aerosol source was changed and the NAM_CO site was closer to dust source after 30 April. This phenomenon had continued until 2 May at NAM_CO and Lhasa, and 1 May at QOMS_CAS. In the last one or two days, the dust decreased while the BC and OC obviously increased, which could cause the mixture of different aerosols.

Then, how was the spatial aerosol loading around the TP and the vertical feature of aerosol transported to the TP? Figure 15 shows the MODIS_AOD and 72-hour back trajectories at Lhasa (the first row), the CALIOP-derived vertical profile of total attenuated backscatter at 532 nm (the second row), and the vertical feature mask of aerosol (the third row) on 28 April, 1 May, and 3 May during the case study period. The MODIS_AOD showed high values in the south (South Asia) and north (Taklimakan Desert) on the three days. The high values in South Asia were caused by anthropogenic aerosols (such as biomass burning) or dust polluted by anthropogenic aerosols, while the high MODIS_AOD in the Taklimakan Desert resulted from dust. The values and areas of the high MODIS_AOD in South Asia and Taklimakan Desert on 1 May and 3 May were higher and larger than those on April 28. The back-trajectories ended at Lhasa on the three days were different. On 28 April, the air flows originated from the southwest (South Asia region). However, the air masses on 1 and 3 May were from the northwest (Taklimakan Desert).

The CALIPSO ground tracks across the TP and through South Asia and the Taklimakan Desert were chosen to show the aerosol transport to the TP sites. On 28 April, the Level-1 attenuated backscatter at 532 nm derived from CALIOP (the second row) showed apparent aerosol layers in the southern area (Bhutan and northeast India) and this aerosol layer even extended to an altitude of ~10 km over the TP along the southern slope of the TP. On 1 May, the CALIOP attenuated backscatter not only showed the deep aerosol layers south of the TP but also showed stronger aerosol layers north of the TP (Taklimakan Desert area). Besides, the north aerosol layers also climbed into the air over the TP, but not as high as the southern aerosol layer. On 3 May, there were also aerosol layers in the south and north of the TP and that were both transported to above the TP, but the aerosol

loading over the TP was lower than that on 28 April and 1 May (the values of attenuated backscatter on 3 May was lower), which corresponded to the lower CE318_AOD on this day than those on 28 April and 1 May at the three TP sites (Figure 13).

The vertical feature mask of the aerosol from CALIOP (the third row) showed the aerosol types on the three days. On 28 April, the aerosol layer in the north (~ 35°N) and above the TP was mainly the smoke aerosol and was even near to 10 km. The back trajectories ended at Lhasa also showed that the southern airflow brought the smoke aerosol and polluted dust from South Asia to the centre of the TP. On 1 May, the aerosol layer on the southern slope of the TP was also smoke aerosol and polluted dust, while the aerosol layers in the northern TP and above the TP were almost all dust aerosol, which could be explained by the northwest airflows carrying the dust aerosol from the Taklimakan Desert. These may be the result of the lower EAE values at Lhasa and NAM_CO than that at QOMS_CAM (Figure 13). On 3 May, the aerosol type above the central TP and the southern TP was occupied by polluted dust aerosol, and the EAE at NAM_CO and QOMS_CAM also showed a slight increase on 3 May. These results agree with the aerosol simulation from GEOS-Chem. Jia et al. (2015) has shown that the dust from India polluted by anthropogenic aerosols can be transported to the TP, but the back trajectories on 1 and 3 May illustrated that the airflows that ended at Lhasa were from the north or northwest rather than the south, indicating that the polluted dust over the TP on 3 May was more likely the mixing result of dust and smoke aerosol. In addition, the lengths of the back trajectories (especially the back trajectories at 10 m and 500 m above ground level) on 1 May showed that the airflows moved slowly, which allowed the possibility of aerosol mixture over the TP. The observations and model simulations illustrated the following scene: first, the smoke aerosol in South Asia was lifted up to 10 km, contaminating the TP sites, and transported to the centre of the TP; then, the dust from the Taklimakan Desert could climb the north slope of the TP and be transported to the TP; finally, the dust and smoke aerosol over the TP were mixed. This case of aerosol pollution shows that the anthropogenic aerosols (mainly smoke) in South Asia and dust in the Taklimakan Desert could be transported to the centre of the TP and they both even can cause mixed aerosol pollution above the TP. The past case studies of aerosol transport to the TP are mostly individual dust or smoke aerosol, while this case of aerosol pollution over the TP showed the mixing pollution during the last two days of the case period.

**6. Conclusion**

The long-term spatiotemporal variations in the aerosol optical properties and the impacts of the long-range aerosol transport over the TP were analysed by using a combination of ground-based and satellite remote sensing aerosol products as well as model simulations. The major conclusions are drawn as follows:

(1) The annual CE318_AOD at most TP sites showed increasing trends (0−0.013/year) during the past decade. Increasing tendencies in the annual averaged EAE were also found at most TP sites. Spatially, the MODIS_AOD showed negative trends in the northwest edge close to the Taklimakan Desert and the east of Qaidam Basin and slightly positive trends in most of the other areas of the TP.

(2) Different aerosol types and sources contributed to the high aerosol loading at the five sites: dust was dominant in Lhasa, Mt_WLG and Muztagh with sources in the Taklimakan Desert, but fine aerosol pollution was dominant at NAM_CO and QOMS_CAS with transport from South Asia.

(3) A case of smoke followed by dust pollution at Lhasa, NAM_CO and QOMS_CAS during 28 April−3 May 2016 showed that the smoke aerosol in South Asia was first uplifted to 10 km and transported to the centre of TP. Then, the dust from the Taklimakan Desert could climb the northern slope of the TP and be transported to the TP, allowing the dust and smoke aerosol over the TP to mix.

There are some limitations to this study. First, ground-based remote sensing and MODIS_AOD products may have had missing data due to clouds interference. Second, only half of a year of observations at the Muztagh_Ata station may not be sufficient to fully reveal pollution days in the northwest TP region, which could have affected the statistics to some extent. More long-term in situ observations are needed in the TP. However, due to the remoteness and challenging weather conditions over the plateau, maintaining long-term in situ observation stations over the TP is very difficult. The numerical model simulation is more practically feasible to study the aerosol properties over the TP, but the model accuracy is far from ideal over the TP. Thus, long-term numerical model simulation coupled with satellite observations and intensive short-term field campaigns should be used to analyse the aerosol properties over the TP in the future.

Data availability:
The four sites (Mt_WLG, Muztagh_Ata, NAM_CO and QOMS_CAS) data are available from AERONET website (https://aeronet.gsfc.nasa.gov/). The dataset of Lhasa used in the study can be requested by contacting the corresponding author. The MODIS aerosol products are available from http://ladsweb.nascom.nasa.gov. The HYSPLIT model and meteorological fields' data can be from https://www.arl.noaa.gov/hysplit/. The CALIPSO data are from https://eosweb.larc.nasa.gov. GEOS-Chem model code and share data can be obtained from http://wiki.seas.harvard.edu/geos-chem.

Competing interests.
The authors declare that they have no conflict of interest.

Author contribution:
All authors help to shape the ideas and review this manuscript. JZ, XX and HC designed, and wrote the manuscript; JZ, XX, HC, JW help to analyze the data; HC, XZ, SK and ZC carried out the sunphotometer observations; JW, ZC, SK, TZ, XY, and YZ provided constructive comments on this study.

**Acknowledgments**
This research was supported by the National Science Fund for Distinguished Young Scholars (41825011), the National Key R & D Program Pilot Projects of China (2016YFA0601901 and 2016YFC0203304), the National Natural Science Foundation of China (41761144056, 41975161), the Strategic Priority Research Program of Chinese Academy of Sciences (XDA20040500), the Natural Science Foundation of Jiangsu Province (BK20170943), the Open fund by the Key Laboratory for Middle Atmosphere and Global Environment Observation (LAGEO) / Institute of Atmospheric Physics, and LAC/CMA (2018B02).

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

**Figure captions**

Figure 1. Topography of the Tibetan Plateau (TP) and the five CE318 stations located in the TP (Lhasa, Mt_WLG, Mutztagh_Ata, NAM_CO, and QOMS_CAS).

Figure 2. Box plots of the monthly aerosol optical depth (AOD) and extinction Ångstrom exponent (EAE) from CE318 at the five sites located on the Tibetan Plateau, i.e., Lhasa, Mt_WLG, Muztagh_Alt, NAM_CO, and QOMS_CAS. In each box, the red-line in the centre is the median and the lower and upper limits are the first and the third quartiles, respectively. The lines extending vertically from the box indicate the spread of the distribution with the length being 1.5 times the difference between the first and the third quartiles. The asterisk symbols indicate the geometric means in each month. The annual mean values and standard errors are also shown in each subgraph.

Figure 3. Seasonal variation in aerosol size distribution at the five sites located in Tibetan Plateau.

Figure 4. Annual averages of and trends in AOD and EAE from CE318 at four sites located in Tibetan Plateau.

Figure 5. Comparisons of the 550 nm AOD measured by the CE318 instrument (CE318_AOD) over Tibetan Plateau stations with the MODIS retrieval Deep-Blue/Dark-Target combined AOD of 10 km spatial resolutions (MODIS_AOD). The statistical parameters in this figure include the number of matchup data (N), the slope and intercept at the y-axis of linear regression (read line), the mean bias (MB), root mean squared error (RMSE), correlation coefficient (R), and the percentage of data within the expected error 0.05+0.15AOD (%EE) which is used as the MODIS AOD expected uncertainty over land (green lines).

Figure 6. Spatial distribution of MODIS_AOD at 550 nm over the Tibetan Plateau (only the altitude > 3000 m) during 2006-2017. The color-filled circles are the CE318 observation AOD averages at TP sites.

Figure 7. The seasonal departure of MODIS_AOD over the Tibetan Plateau (altitude > 3000 m).

Figure 8. Trend in the MODIS_AOD at 550 nm during 2006-2017.

Figure 9. Trends in the MODIS_AOD at 550 nm during 2006-2017 in each season.

Figure 10. AOD vs EAE (only CE318_AOD at 440 nm > 0.4 was considered) observed by CE318 at the five sites on the Tibetan Plateau.

Figure 11. Aerosol size distribution binned by CE318_AOD at the five sites on the Tibetan Plateau.

Figure 12. The back-trajectories ended at the five sites (10 m above ground level) on the Tibetan Plateau overlaid with the mean MODIS_AOD at 550 nm on the high aerosol loading day observed by ground-based CE318 (CE318_AOD >0.4). Red stands for EAE >1.0, black for EAE < 0.5, and green for EAE within 0.5-1.0.

Figure 13. CE318 observed daily AOD and EAE during 27April, 2016 – 3 May, 2016 at Lhasa, NAM_CO and QOMS_CAS.

Figure 14. The model evaluation of GEOS-Chem model simulated the daily average AOD (Model_AOD) by using the CE318 observed daily AOD (CE318_AOD) at 550 nm, and the model results of the ratios of dust or organic carbon (OC) and black carbon (BC) aerosol to the total AOD

during 27April, 2016 –3 May, 2016 at Lhasa, NAM_CO and QOMS_CAS. The statistical parameters used in the modal evaluation are the same as those in Figure 5.

Figure 15. The MODIS_AOD at 550 nm and 72-hour back trajectories ended at Lhasa at three heights above the ground level (10 m in black, 500 m in red and 1000 m in blue lines) (the first row); the CALIOP-derived vertical profile of total attenuated backscatter at 532 nm (the second row); and the vertical feature mask of aerosol on 28 April, 1 May, and 3 May, 2016 over the ground track (green lines) shown in the first row (the third row).

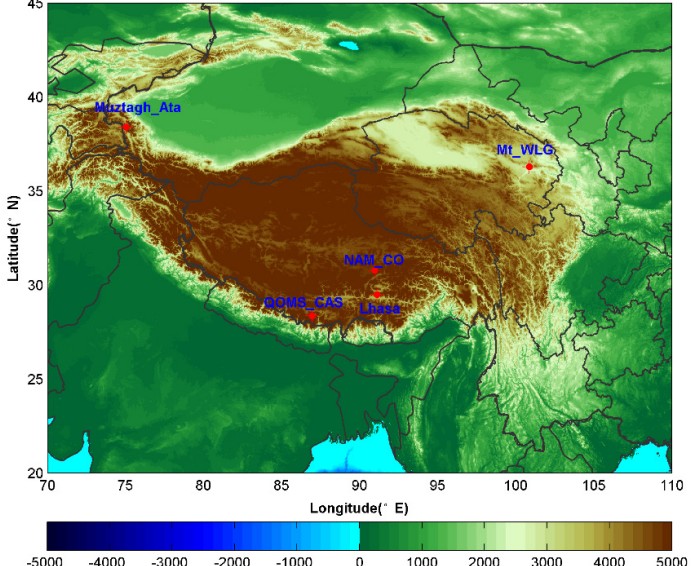

Figure 1. Topography of the Tibetan Plateau (TP) and the five CE318 stations located in the TP (Lhasa,
Mt_WLG, Mutztagh_Ata, NAM_CO, and QOMS_CAS).

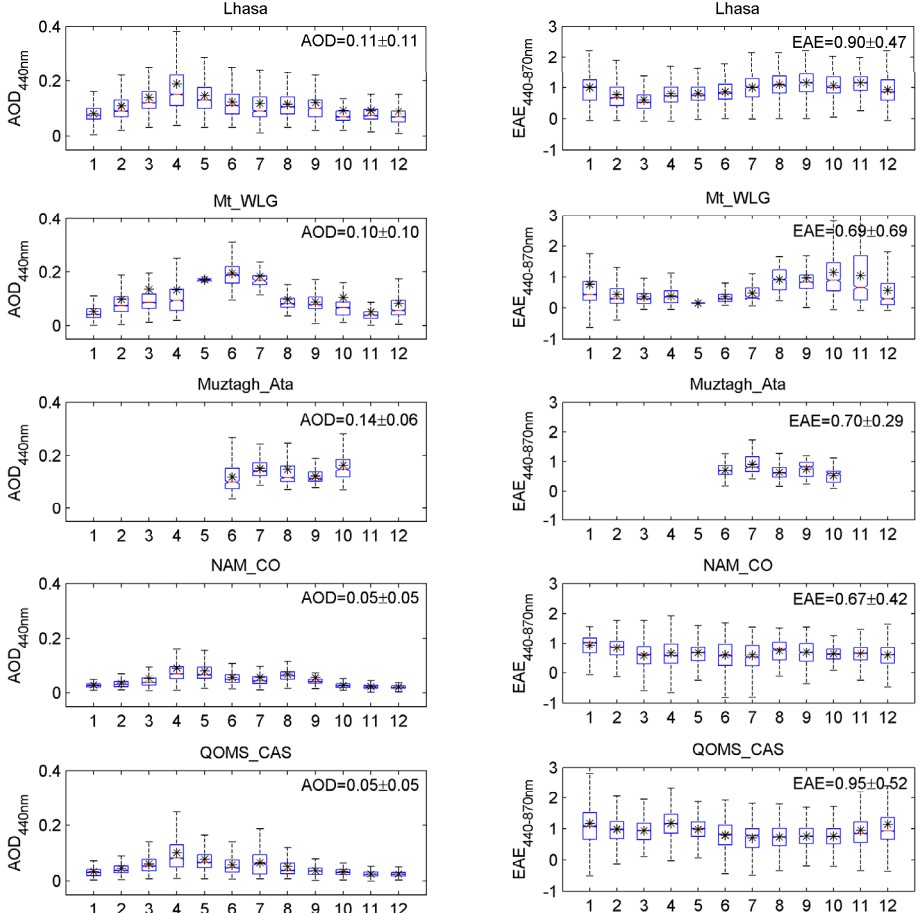

Figure 2. Box plots of the monthly aerosol optical depth (AOD) and extinction Ångstrom exponent (EAE) from CE318 at the five sites located on the Tibetan Plateau, i.e., Lhasa, Mt_WLG, Muztagh_Alt, NAM_CO, and QOMS_CAS. In each box, the red-line in the centre is the median and the lower and upper limits are the first and the third quartiles, respectively. The lines extending vertically from the box indicate the spread of the distribution with the length being 1.5 times the difference between the first and the third quartiles. The asterisk symbols indicate the geometric means in each month. The annual mean values and standard errors are also shown in each subgraph.

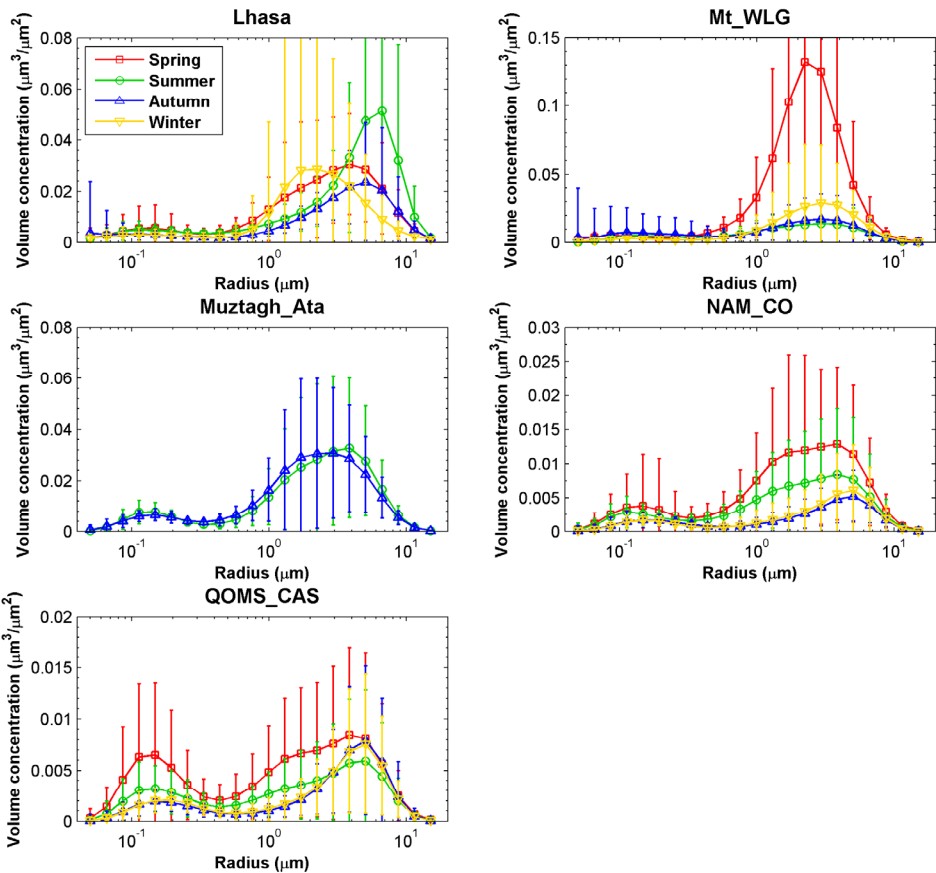

2    Figure 3. Seasonal variation in aerosol size distribution at the five sites located in Tibetan Plateau.

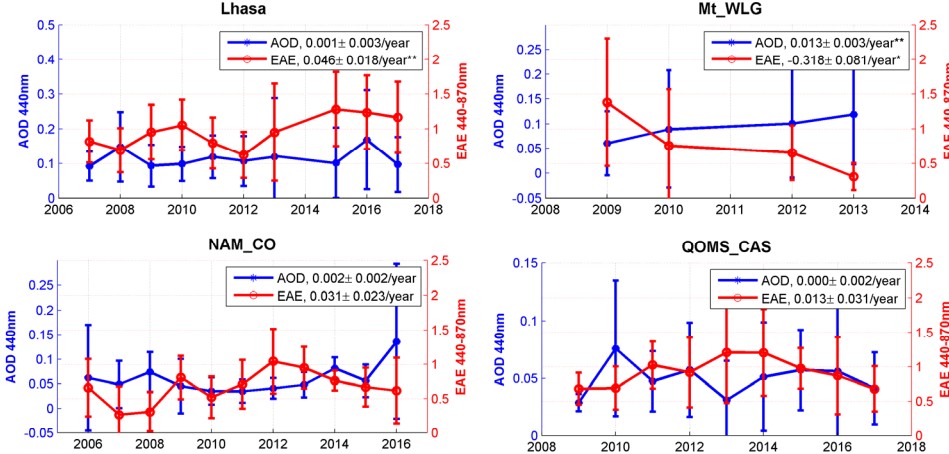

2    Figure 4. Annual averages of and trends in AOD and EAE from CE318 at four sites located in Tibetan

3    Plateau. * stands for 90% significance, and ** represents 95% significance.

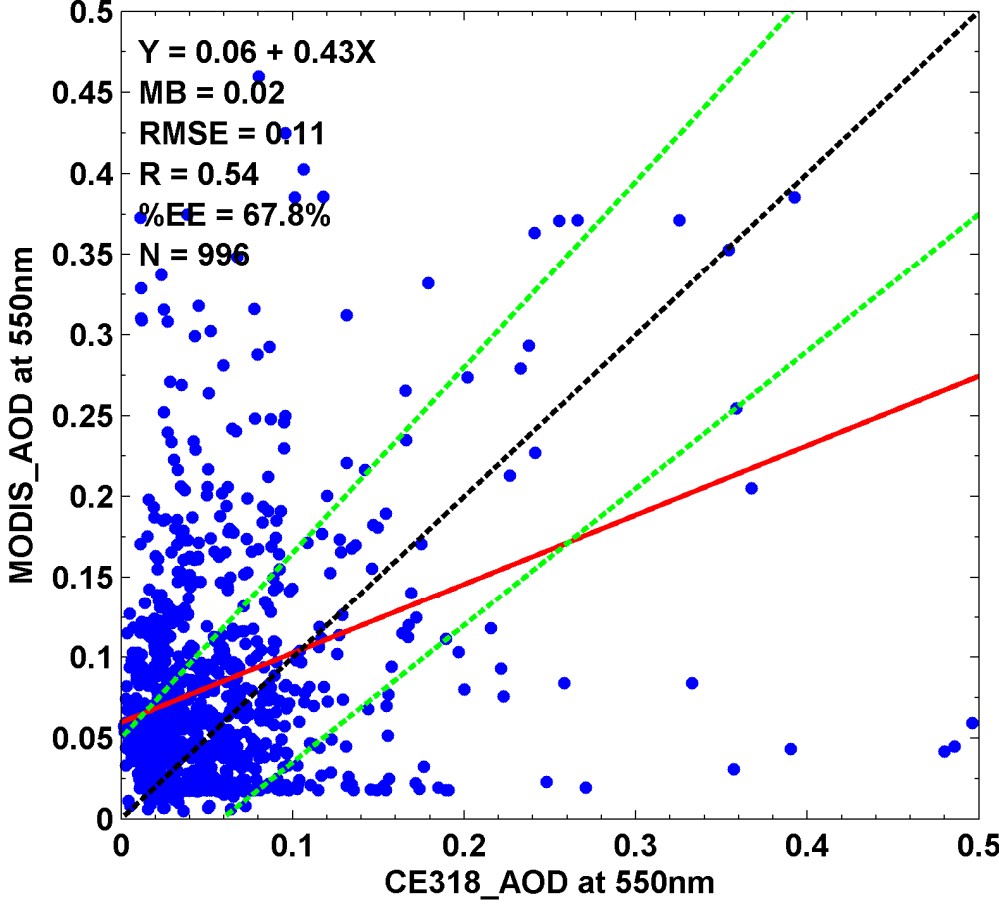

3   Figure 5. Comparisons of the 550 nm AOD measured by the CE318 instrument (CE318_AOD) over

4   Tibetan Plateau stations with the MODIS retrieval Deep-Blue/Dark-Target combined AOD of 10 km

5   spatial resolutions (MODIS_AOD). The statistical parameters in this figure include the number of

6   matchup data (N), the slope and intercept at the y-axis of linear regression (read line), the mean bias

7   (MB), root mean squared error (RMSE), correlation coefficient (R), and the percentage of data within

8   the expected error 0.05+0.15AOD (%EE) which is used as the MODIS AOD expected uncertainty over

9   land (green lines).

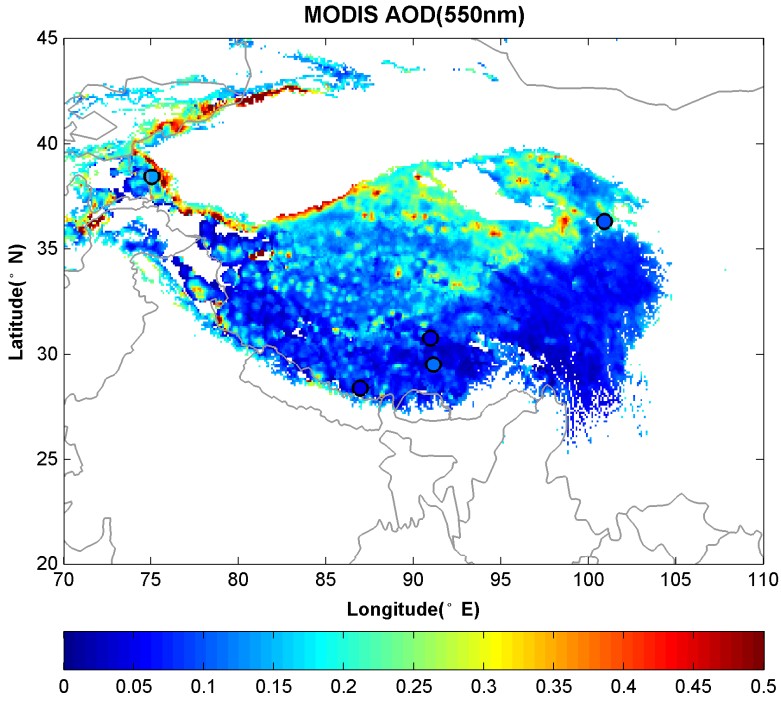

2  Figure 6. Spatial distribution of MODIS_AOD at 550 nm over the Tibetan Plateau (only the altitude >

3  3000 m) during 2006-2017. The color-filled circles are the CE318 observation AOD averages at TP sites.

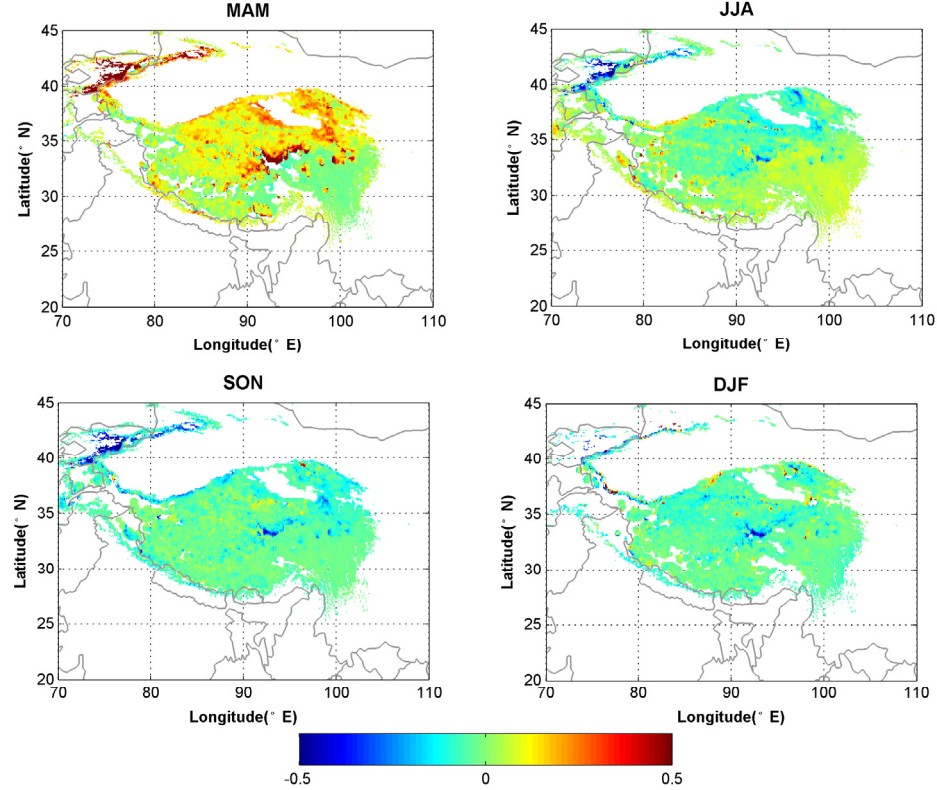

Figure 7. The seasonal departure of MODIS_AOD over the Tibetan Plateau (altitude > 3000 m).

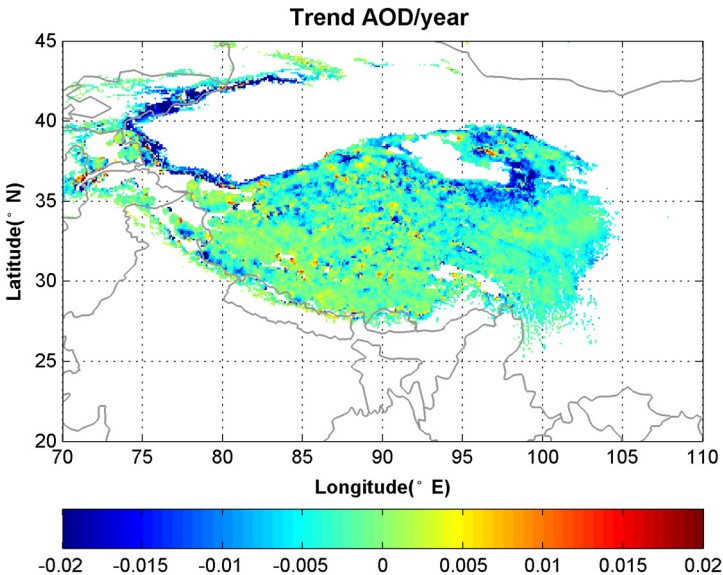

Figure 8. Trend in the MODIS_AOD at 550 nm during 2006-2017.

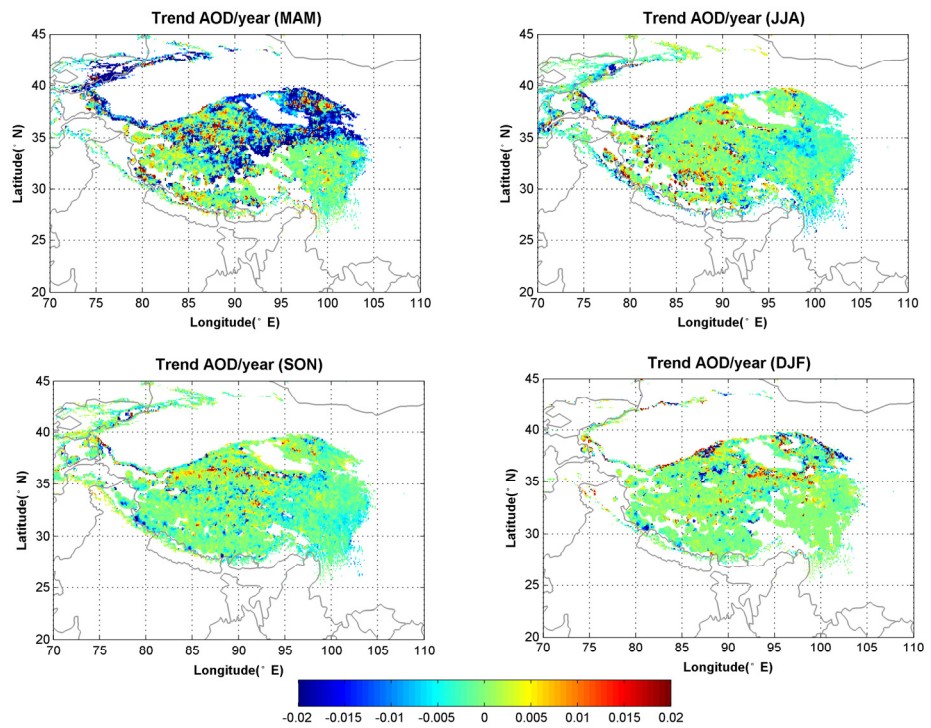

Figure 9. Trends in the MODIS_AOD at 550 nm during 2006-2017 in each season.

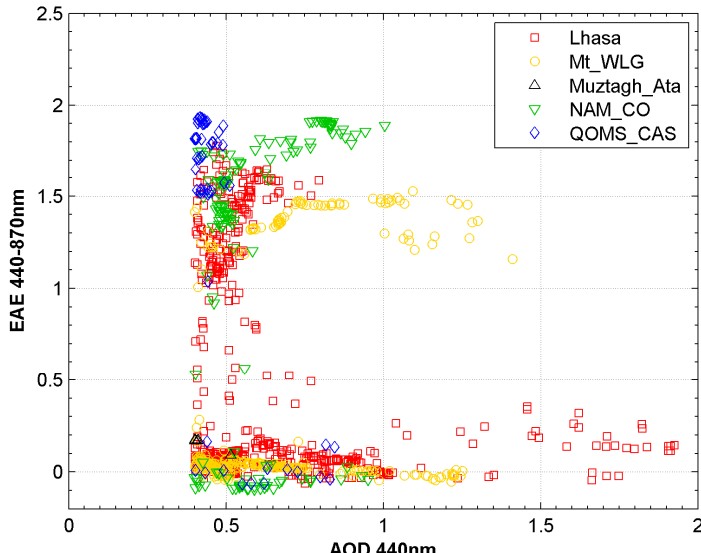

2 Figure 10. AOD vs EAE (only CE318_AOD at 440 nm > 0.4 was considered) observed by CE318 at
3 the five sites on the Tibetan Plateau.

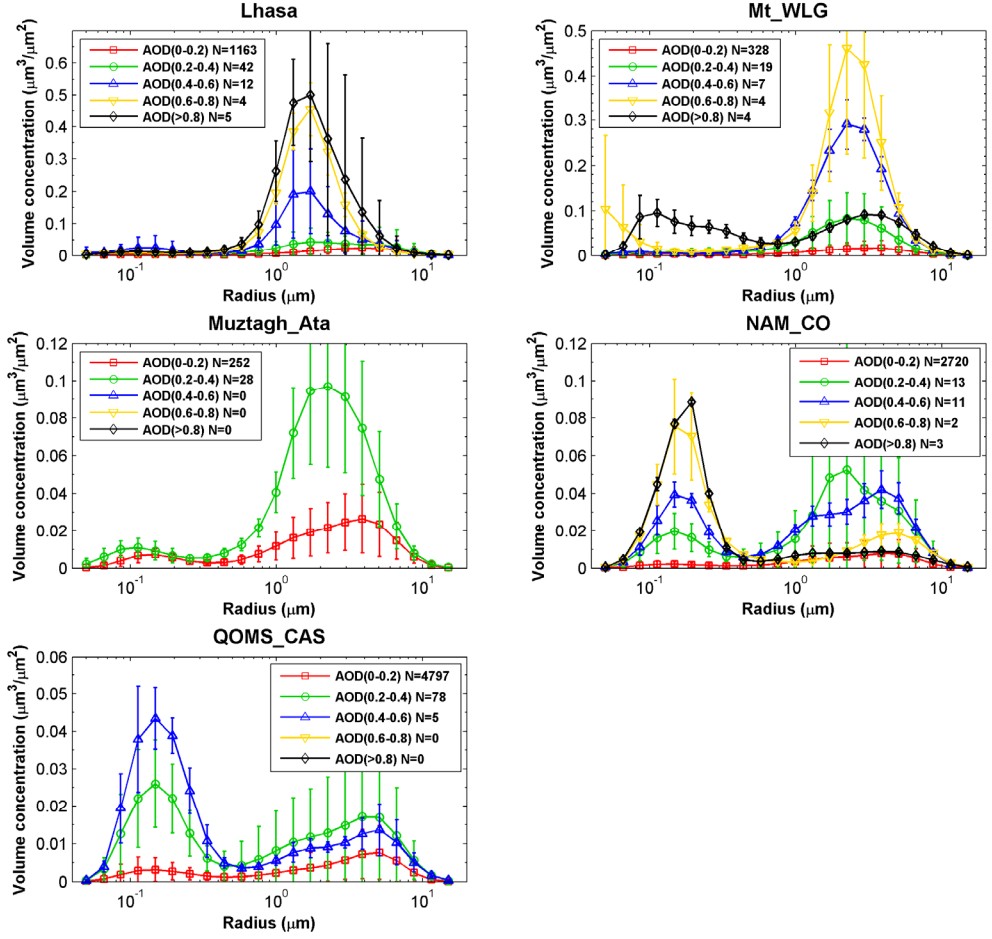

2    Figure 11. Aerosol size distribution binned by CE318_AOD at the five sites on the Tibetan Plateau.

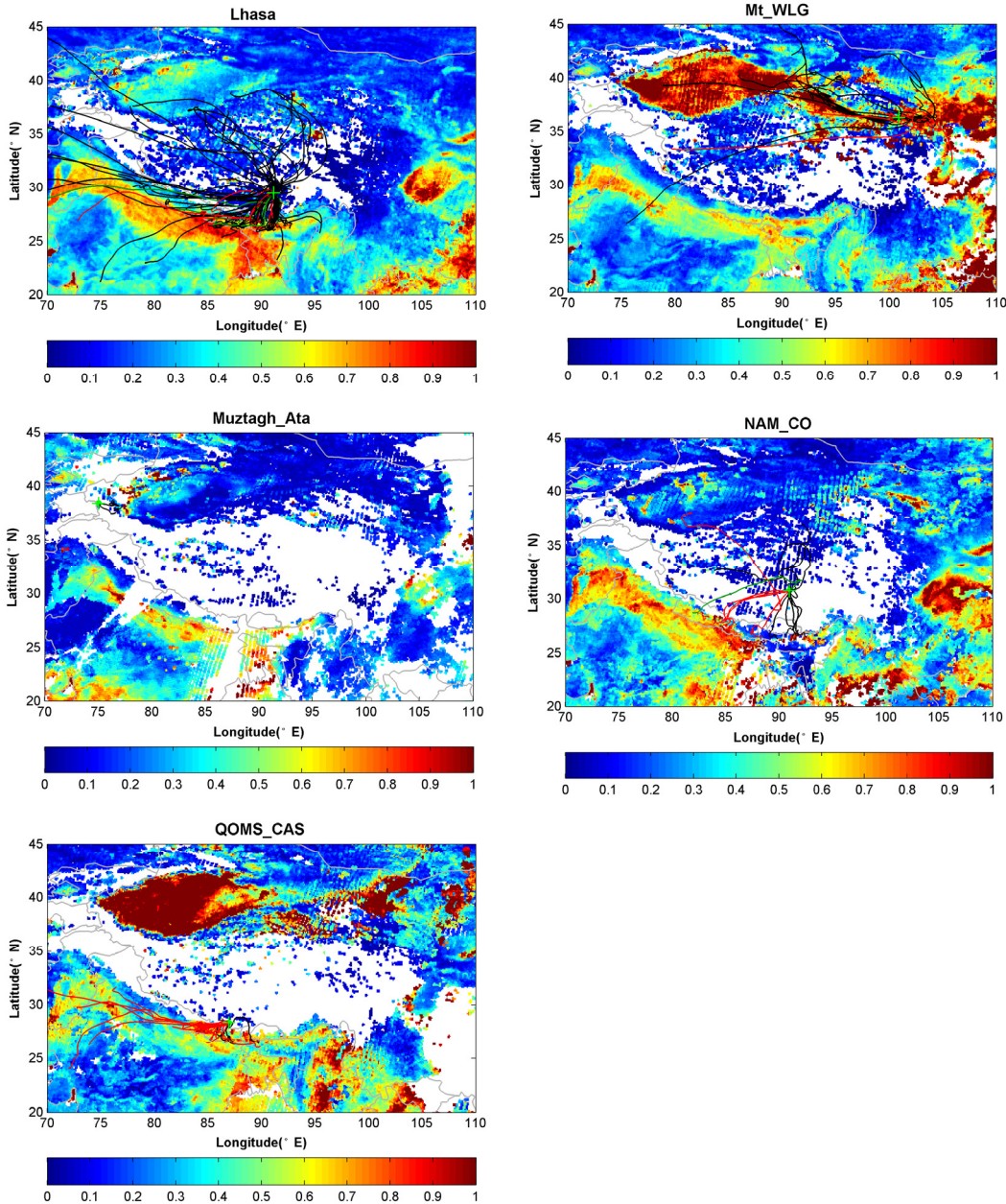

Figure 12. The back-trajectories ended at the five sites (10 m above ground level) on the Tibetan Plateau overlaid with the mean MODIS_AOD at 550 nm on the high aerosol loading day observed by ground-based CE318 (CE318_AOD >0.4). Red stands for EAE >1.0, black for EAE < 0.5, and green for EAE within 0.5-1.0.

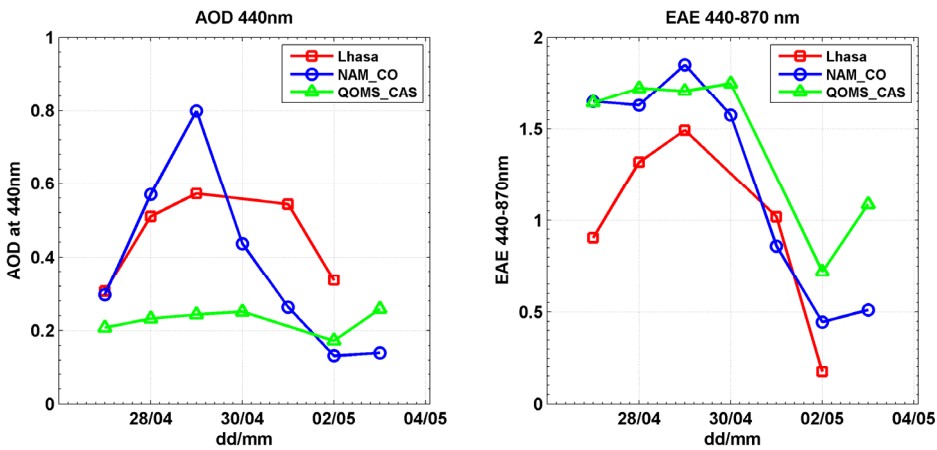

2    Figure 13. CE318 observed daily AOD and EAE during 27April, 2016 – 3 May, 2016 at Lhasa, NAM_CO

3    and QOMS_CAS.

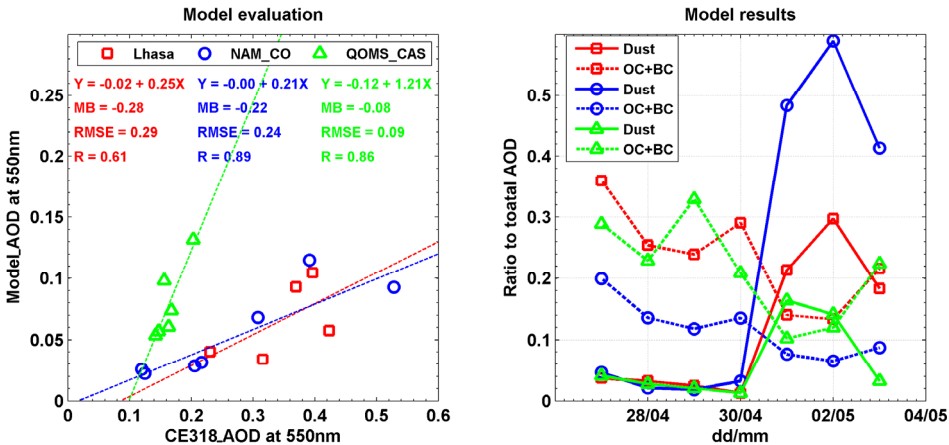

2  Figure 14. The model evaluation of GEOS-Chem model simulated the daily average AOD (Model_AOD)

3  by using the CE318 observed daily AOD (CE318_AOD) at 550 nm, and the model results of the ratios

4  of dust or organic carbon (OC) and black carbon (BC) aerosol to the total AOD during 27April, 2016 –3

5  May, 2016 at Lhasa, NAM_CO and QOMS_CAS. The statistical parameters used in the modal evaluation

6  are the same as those in Figure 5.

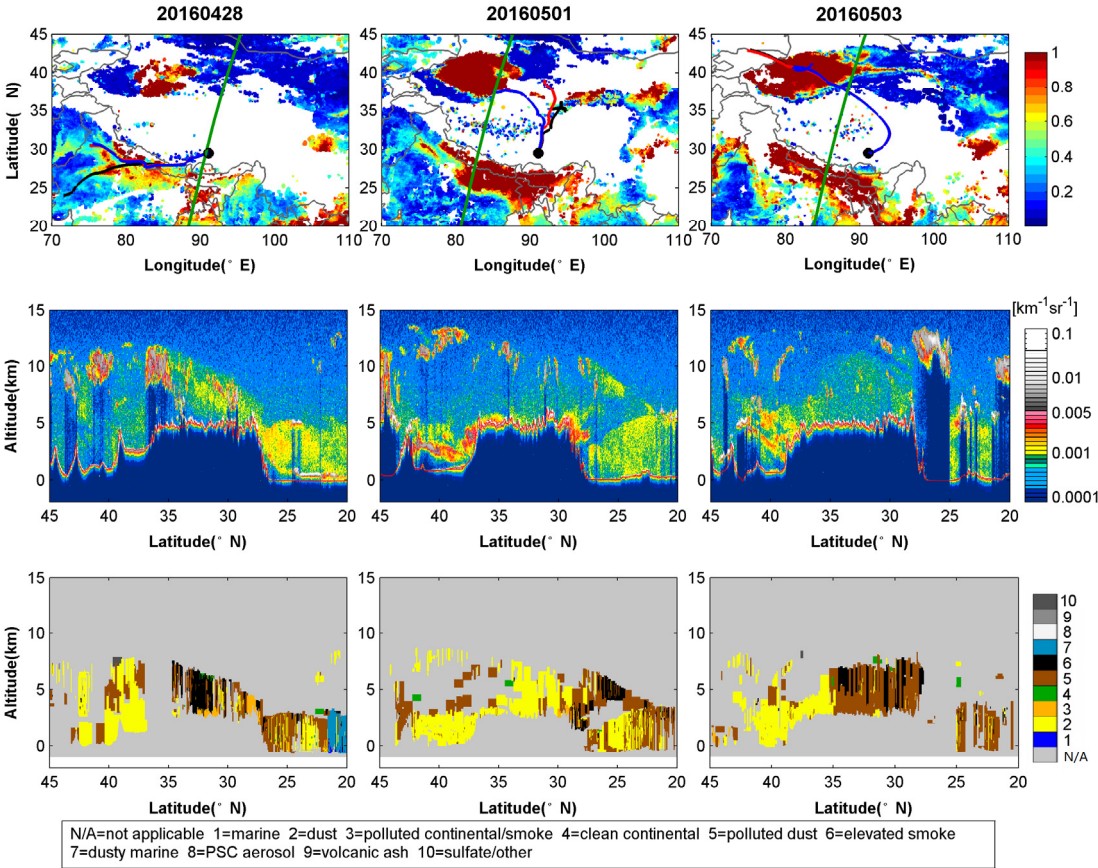

Figure 15. The MODIS_AOD at 550 nm and 72-hour back trajectories ended at Lhasa at three heights above the ground level (10 m in black, 500 m in red and 1000 m in blue lines) (the first row); the CALIOP-derived vertical profile of total attenuated backscatter at 532 nm (the second row); and the vertical feature mask of aerosol on 28 April, 1 May, and 3 May, 2016 over the ground track (green lines) shown in the first row (the third row).

1    Table 1. Site location and description.

| Site name | Lat(° N) | Lon(° E) | Site description, observation days and period |
|---|---|---|---|
| Lhasa | 29.50 | 91.13 | Urban station on the Tibetan Plateau, 3648 m a.s.l., 1554 days, 2007.05~2017.12 |
| Mt_WLG | 36.28 | 100.90 | Mountain, 3816 m a.s.l., 314 days, 2009.09~2013.07 |
| Muztagh_Ata | 38.41 | 75.04 | Mountain, 3674 m a.s.l., 84 days, 2011.06~2011.10 |
| NAM_CO | 30.77 | 90.96 | Mountain, 4740 m a.s.l., 1061 days, 2006.08~2016.08 |
| QOMS_CAS | 28.36 | 86.95 | Mountain, 4276 m a.s.l., 1623 days, 2009.10~2017.11 |

1 Table 2. Seasonal aerosol optical depth ($AOD_{440nm}$) and extinction Angstrom exponent ($EAE_{440-870nm}$)

2 from CE318 at the five sites in the TP.

| Site | AOD | | | | EAE | | | |
|------|-----|-----|-----|-----|-----|-----|-----|-----|
| | MAM | JJA | SON | DJF | MAM | JJA | SON | DJF |
| **Lhasa** | 0.16+0.10 | 0.12+0.08 | 0.10+0.18 | 0.09+0.08 | 0.72+0.37 | 0.97+0.40 | 1.11+0.38 | 0.91+0.52 |
| **Mt_WLG** | 0.13+0.16 | 0.14+0.07 | 0.08+0.11 | 0.08+0.07 | 0.37+0.38 | 0.65+0.40 | 1.04+0.80 | 0.58+0.69 |
| **Muztagh_Ata** | NaN | 0.14+0.06 | 0.14+0.05 | NaN | NaN | 0.73+0.30 | 0.64+0.27 | NaN |
| **NAM_CO** | 0.07+0.07 | 0.06+0.04 | 0.03+0.05 | 0.03+0.01 | 0.63+0.44 | 0.62+0.45 | 0.65+0.32 | 0.78+0.43 |
| **QOMS_CAS** | 0.08+0.06 | 0.06+0.04 | 0.03+0.01 | 0.03+0.02 | 1.04+0.38 | 0.76+0.43 | 0.85+0.51 | 1.10+0.67 |

Table 3. The percentages of EAE <0.5, 0.5-1.0, and >1.0 for high AOD observations at the five sites.

| Site | N of AOD>0.4 | % EAE<0.5/N | % 0.5<EAE<1.0/N | % EAE>1.0/N |
|---|---|---|---|---|
| Lhasa | 655 | 60.6 | 3.4 | 36.0 |
| Mt_WLG | 290 | 73.4 | 0 | 26.6 |
| Muztagh_Ata | 5 | 100 | 0 | 0 |
| NAM_CO | 140 | 27.9 | 2.8 | 69.3 |
| QOMS_CAS | 59 | 23.7 | 0 | 76.3 |