# Peer review of "Spatiotemporal variation of aerosol and potential long-range transport impact over Tibetan Plateau, China"

_Atmospheric Chemistry and Physics, 2019_

## Referee Comment (RC1) · Anonymous Referee #1 · 17 Jun 2019

General comments: This manuscript presents a study of the features of aerosols over the Tibetan Plateau (TP), including the distribution of AOD and Extinction Ångstrom exponent, the types and sources of aerosols. The utilization of sunphotometer measurements (CE318) is effective, which is significant to provide evidence of aerosol properties over the TP. However, some major revisions including content organization are needed. Moreover, improvement in English is needed before the paper can be accepted for publication. Therefore, I recommend publication after the authors address the following issues.

Major comments: 1.The combination of case and long-term study, ground-based and

satellite observation analysis together with the model simulation including two models need reorganized according to the scientific goal in this study. 2.The reliability of CE318 observation should be described in Section 2.2.1. Though the authors illustrated the errors, the situation of instrument calibration should be described here. 3.What is the reason of "The CE318 observed AOD larger than 0.4 at each site is considered as the aerosol pollution 12 over TP"? An appropriate reference is needed, or the background AOD should be provided. 4.What is the role of GEOS-Chem model? According to the role of model, in the methodology, the details of model description should be shown separately. 5.In Section 3.1, the wavelength of AOD analyzed here should be given. Moreover, the authors analyzed the trend of AOD in Section 3.1, a significance check is needed. 6.What is the purpose of using CLIPSO observation data? 7.What is the relationship between the ground-based and satellite observations? Since the authors have the valuable ground-based data, an evaluation of satellite observation, including MODIS and CALIPSO, can be performed, which is a good basis to get the spatial variation of aerosol properties in Section 3.2. 8.Page 5 Line 43 and 44, the authors think the positive trend of AOD and EAE at most sites over TP is caused by the addition of fine mode aerosol mainly from the anthropogenic impact. However, dust aerosols transported to the TP over long distances also has a small particle radius, causing similar changes. Thus, the authors should also take it into consideration. 9.Figure 3 contains a lot of information, the authors need to indicate whether the values in the paper are the average, median or otherwise. 10.The authors mainly consider the anthropogenic aerosols in Southeast Asia, however, according to some research (e.g., Jia et al., AE, 2015), dust storm also occurs in the Indian peninsula. Can the authors separately estimate the contribution of anthropogenic aerosol and dust aerosol transported to the TP from Southeast Asia? Can the GEOS-Chem gives such evidence? Minor comments: 1.Page 3, Line9-10, what is the meaning of "large scale"? Spatial scale or temporal scale? The sentence need be illustrated clearly. 2.Page 3 Line 10, "satellite remote-sensing method (Li et." should be "satellite remote-sensing method (Li et.", in which a space is needed between "method" and "(". In the whole manuscript, such writing

problem should be paid attention, for example, Page 3, Line 20, there should be a space between "2007;" and "Xia", etc. 3.Page 3, Line 27, there is mistake in grammar in sentence "there is an urgent need to......". 4.Page 4, Line 3, "2.1 site" should be "2.1 Site". 5.Page 4, Line 6, there is mistake in grammar in sentence "site where can suffer from the local anthropogenic emissions". 6.Before the unit, there need a space, for example, Page 4, Line 34, "2330km". 7.Page 5 Line 41 and 42, 'Mt_WLG sites' should be 'Mt_WLG site'.

---

## Referee Comment (RC2) · Anonymous Referee #2 · 9 Aug 2019

General comments: Tibetan Plateau (TP) plays a very important role in East Asian climate. Perturbation in thermodynamic fields of the Qinghai-Xizang Plateau by anthropogenic or natural aerosols might induce substantial regional climate changes and serious air pollutions. However, the variations of aerosols in TP region are less known compared with those in East or South Asian regions. This study investigates the characteristics and potential sources of aerosols in TP based on ground-based and satellite observations as well as numerical models. The results are interesting and they may help us better understanding the temporal and spatial variations of the aerosols in TP and subsequently the aerosol climate effects in Asian region. The topic of this study is novel to some degrees. And the paper has a potential for publication in the journal

after revisions. Comments: 1. Introduction should be re-organized to a degree to make it more readable and more clearly. 2. The authors should make some comparisons of aerosol optical properties which derive from different platforms when investigating the temporal and spatial variations of aerosols in TP region. 3. A more detailed description on the accuracy of each type of platform data is needed. Does MODIS products accurate enough in bright surface (such as in desert region in TP)? 4. Validation of GEOS-Chem is need. The authors should compare the simulated aerosols with the observations. 5. How frequency of aerosol pollutions in Qinghai-Tibet Plateau based on your study? 6. A deeper discussion is needed in Results section, such as make some comparisons or summaries from similar studies. 7. Conclusions should be shortened and more concise. 8. English should be improved substantially throughout the whole manuscript.
* * *

---

## Author Comment (AC1) · 27 Sep 2019

The comment was uploaded in the form of a supplement:
https://www.atmos-chem-phys-discuss.net/acp-2019-444/acp-2019-444-AC1-supplement.pdf

---

## Author Response (AR1)

**Spatiotemporal variation of aerosol and potential long-range transport impact over Tibetan Plateau, China**

**Jun Zhu et al.**

We appreciate the reviewers for their constructive comments and suggestions. The manuscript has been revised accordingly. Our point-by-point responses to the comments are presented below. The comments are in black, followed by responses in blue and revised manuscript in *blue* with changes marked by underline.

**Response to Comments of Reviewer #1**

**General comments:** This manuscript presents a study of the features of aerosols over the Tibetan Plateau (TP), including the distribution of AOD and Extinction Ångstrom exponent, the types and sources of aerosols. The utilization of sunphotometer measurements (CE318) is effective, which is significant to provide evidence of aerosol properties over the TP. However, some major revisions including content organization are needed. Moreover, improvement in English is needed before the paper can be accepted for publication. Therefore, I recommend publication after the authors address the following issues.

Response: We are very grateful for your important and constructive comments and suggestions. Some major revisions have been made carefully according to the comments and suggestions of this manuscript. Moreover, the grammar in the paper has been carefully checked and the language of this manuscript has been edited by native English speakers.

**Major comments:**

1.The combination of case and long-term study, ground-based and satellite observation analysis together with the model simulation including two models need reorganized according to the scientific goal in this study.

Response: We have tried our best to reorganize the content according to the reviewer's suggestion. Some large modifications and detail adjustments have been made in the revised version.

The content of section 3.1 is reorganized (the adjusted order is from monthly to seasonal and then to annual variations) and the titles of section 3.1 and 3.2 have been corrected as "*Aerosol properties observed by the CE318 instruments*" and "*Aerosol properties from MODIS*", respectively.

And, at the first of section 3.2, a statement of the relationship of ground-based to satellite observation ("*Ground-based observations can offer accurate aerosol optical properties at point locations but lack spatial coverage. The MODIS aerosol product can provide the spatial variation of AOD over the TP.*") has been added to connect the section 3.1 ground-based measurement and section 3.2 satellite observation.

In order to connect the case to the above, a transition has been added at the first of section 5. "*The aerosol long-range transport can cause the aerosol pollution and affect the long-term aerosol variation over the TP. In addition, the dominant aerosol type may change at the TP sites during a case of aerosol transport. Thus…*"

As for last paragraph of case analysis, the model simulation and HYSPLIT back trajectories have been combined with the ground and satellite observations to show the aerosol transport and mixture over the TP.

2.The reliability of CE318 observation should be described in Section 2.2.1. Though the authors illustrated the errors, the situation of instrument calibration should be described here.

Response: In order to verify the accuracy and reliability, we have added the data retrieval references ("*Dubovik and King, 2000; Dubovik et al., 2006*") and the instrument calibration in section 2.2.1, as followed:

"*The instruments were periodically calibrated using the Langley method at AERONET global calibration sites (the Izaña, Spain or the Mauna Loa, USA) or using the inter-comparison calibration method at the Beijing-CAMS site (Che et al., 2015). The cloud-screened and quality-controlled data of AOD, Extinction Ångstrom exponent (EAE), and aerosol volume size distribution (dV(r)/dlnr) are used in this work (Giles et al., 2019).*"

3.What is the reason of "The CE318 observed AOD larger than 0.4 at each site is considered as the aerosol pollution over TP"? An appropriate reference is needed, or the background AOD should be provided.

Response: The figure 2 showed the annual mean values of AOD at 440nm at the five Tibetan Plateau sites are less than 0.14. Xia et al., (2015) and Cong et al., (2009) have showed the mean AODs observed by CE318 instruments at TP sites were less than 0.11. Thus, the value of 0.4 is larger than the three times the mean value at TP CE318 sunphotometer sites. Besides, the value of 0.4 is normal regarded as high aerosol loading (Eck et al., 2010; Giles et al., 2012). According to this suggestion, the sentence has been changed and the reason is added as followed:

"*The CE318 observed AOD at 440 nm with values larger than 0.4 at each site was specially analysed to study the aerosol properties of the high aerosol loading over the TP. The value of 0.4 was selected because the mean annual values of AOD observed by CE318 instruments at the TP sites were less than ~0.1 in the past studies (Xia et al., 2016; Cong et al., 2009), and this value is normally regarded as the high aerosol loading (Eck et al., 2010; Giles et al., 2012)*"

4.What is the role of GEOS-Chem model? According to the role of model, in the methodology, the details of model description should be shown separately.

Response: The GEOS-Chem model was used to simulate the aerosol variation during the case period. According to this suggestion, a separate paragraph of GEOS-Chem model description has been added as following:

"*The GEOS-Chem chemical transport model (version 11-01) coupled with the online radiative transfer calculations (RRTMG) at 0.5° × 0.667° horizontal resolution over the East Asia domain (Bey et al., 2001; Wang et al., 2004) was used. The model was driving by the Global Modeling and Assimilation Office (GMAO) MERRA-2 meteorology with the temporal resolution of 3 hours for meteorological parameters and 1 hour for surface fields. The simulation type of full chemistry in the troposphere was selected. The implementation of RRTMG in GEOS-Chem was described in Heald et al. (2014). The AOD was calculated according to Martin et al. (2003). The default global anthropogenic emissions were overwritten over East Asia by the MIX inventory from Li et al. (2014). The Global Fire Emission Database (GFED) (van der Werf et al., 2010) has been used to specify emissions from fire. More details on the model and the other emissions data used and the evaluation of AOD in the east and south of the TP were shown in Zhu et al. (2017)*"

5.In Section 3.1, the wavelength of AOD analyzed here should be given. Moreover, the authors analyzed the trend of AOD in Section 3.1, a significance check is needed.

Response: Thanks the reviewer's comments. We have added the wavelength description of this manuscript in section 2.3, i.e., "*In this study, the AOD from the CE318, MODIS, and GEOS-Chem model were used. For convenience, CE318_AOD, MODIS_AOD, and Model_AOD stand for the AOD observed by CE318, MODIS, and the AOD simulated by the GEOS-Chem model, respectively. For CE318_AOD, the 440 nm wavelength is often studied, while MODIS_AOD and Model_AOD generally use the data at 550 nm wavelength. Thus, unless otherwise specified, CE318_AOD, MODIS_AOD, and Model_AOD hereinafter represent the ones at 440 nm, 550 nm, and 550 nm, respectively.*"

According to the reviewer comment, we have added the markers at the site which meet the 90% and 95% significances level in the corresponding figure. "*\* stands for 90% significance and \*\* represents 95% significance.*"

6.What is the purpose of using CALIPSO observation data?

Response: The CALIPSO data were used to show the vertical feature of aerosol (including aerosol profile and aerosol type) during the case period. In revised version, the purpose of using CALIOP has been added in the third paragraph of section 2.3 with MODIS and HYSPLIT back trajectories as followed:

"*The HYSPLIT back trajectories, and the MODIS and CALIOP products were used to show the potential aerosol sources, spatial aerosol loading and the vertical features of the aerosol over the TP during the case period.*"

7.What is the relationship between the ground-based and satellite observations? Since the authors have the valuable ground-based data, an evaluation of satellite observation, including MODIS and CALIPSO, can be performed, which is a good basis to get the spatial variation of aerosol properties in Section 3.2.

Response: Ground-based observation can offer more accurate aerosol optical properties at only one location (point) but lack spatial coverage. Satellite observation can make up for it. Hence, they are complementary. In the section 2.2, we had introduced some references about the evaluations of the satellite data. The simple comparison of mean values between CE318 and MODIS was shown in figure 5 in the original version. According to the reviewers' suggestion, we have added the comparison of MODIS_AOD and CE318_AOD in revised version at section 3.2 as followed:

"*Ground-based observations can offer accurate aerosol optical properties at point locations but lack spatial coverage. The MODIS aerosol product can provide the spatial variation in AOD over the TP. Thus, we evaluated the MODIS_AOD using the ground-based observation CE318_AOD at 550 nm over the TP sites. The CE318_AOD at 550 nm was interpolated from 440 nm, 675 nm, 870 nm and 1020 nm by using an established fitting method from Ångström (1929). The matchup method was that the CE318 data within 1 hour of the MODIS overpass were compared with the MODIS data within a 25 km radius of the ground-based site. The minimum requirement for a matchup was at least 3 pixels from MODIS.*

*Figure 5 shows the results of MODIS_AOD compared to the collocated ground CE318 observations over the TP. There are 996 instantaneous matchups of Terra and Aqua MODIS during the CE318 instrument measurement period at the five TP sites. The MODIS_AOD overestimates the AOD at 550 nm with a positive mean bias of 0.02 and a root mean squared error (RMSE) of 0.11. The RMSE value is lower than that of the North China Plain sites (~0.25) (Bilal et al., 2019). The slope and intercept of the best-fit equation between the MODIS_AOD and CE318_AOD at 550 nm are 0.46 and 0.06, respectively, with a correlation coefficient (R) of 0.54. There are 67.8% of the compared AODs within the expected error envelope of 0.05+0.15AOD (%EE). The R value is lower than that in the global assessment statistics, while the %EE is higher than that in the global evaluation (Bilal and Qiu, 2018). Overall, the results suggest that the MODIS_AOD product can be used to study the aerosol spatial variation over the TP region.*

[Figure]

*Figure 5. Comparisons of the 550 nm AOD measured by the CE318 instrument*
*(CE318_AOD) over Tibetan Plateau stations with the MODIS retrieval Deep-*
*Blue/Dark-Target combined AOD of 10 km spatial resolutions (MODIS_AOD). The*
*statistical parameters in this figure include the number of matchup data (N), the slope*
*and intercept at the y-axis of linear regression (read line), the mean bias (MB), root*
*mean squared error (RMSE), correlation coefficient (R), and the percentage of data*
*within the expected error 0.05+0.15AOD (%EE) which is used as the MODIS AOD*
*expected uncertainty over land (green lines).*"

In this study, MODIS AOD was used to show the spatial variation which can cover the
shortage of CE318 sunphotometer observations. But CALISPO data were only used to
show the vertical feature of aerosol during the case period. Thus, we have added the
evaluation reference of CALIPSO data in section 2.3 as followed:
"*Kumar et al. (2018) have showed that the AOD from CALIOP version 4.10 agreed*
*with the ground-based CE318 observation at a site in the central Himalayas with a*
*correlation > 0.9 and ~ 87 % matchup data were within the expected error.*"

8.Page 5 Line 43 and 44, the authors think the positive trend of AOD and EAE at most
sites over TP is caused by the addition of fine mode aerosol mainly from the
anthropogenic impact. However, dust aerosols transported to the TP over long distances
also has a small particle radius, causing similar changes. Thus, the authors should also
take it into consideration.
Response: Agree with this comment. This sentence has been change as "*Looking at the*
*CE318_AOD and EAE values together, the positive trend of CE318_AOD and the*
*positive trend of EAE in the long term variation at most sites over TP indicates the*
*addition of fine mode aerosol which may be related to the anthropogenic impact or*
*long-distance transport of dust to the TP.*"

9.Figure 3 contains a lot of information, the authors need to indicate whether the values in the paper are the average, median or otherwise.

Response: Thanks for the comments. The values used in the paper are the averages, including monthly, seasonal and annual averages. We have added the statement of monthly and annual means (averages) in the figure caption ("*The asterisk symbols indicate the geometric means in each month. The annual mean values and standard errors are also shown in each subgraph.*") and the corresponding text, such as "*However, the monthly mean CE318_AOD at Mt_WLG is nearly symmetrical...*" in second paragraph in section 3.1 and "*This size distribution explained the relatively low annual averages of EAE...*" at the fourth paragraph in section 3.1 in the revised version.

10.The authors mainly consider the anthropogenic aerosols in Southeast Asia, however, according to some research (e.g., Jia et al., AE,2015), dust storm also occurs in the Indian peninsula. Can the authors separately estimate the contribution of anthropogenic aerosol and dust aerosol transported to the TP from Southeast Asia? Can the GEOS-Chem gives such evidence?

Response: According the reviewer's comments, we have made some modification at the related content. The fourth paragraph of section 5, "*High values in South Asia was caused by biomass burning, while...*" has been corrected as "*The high values in South Asia were caused by anthropogenic aerosols (such as biomass burning) or dust polluted by anthropogenic aerosols...*". Besides, in the last paragraph of section 5, this reference has been added and discussed in this case, as followed:

"*Jia et al. (2015) has shown that the dust from India polluted by anthropogenic aerosols can be transported to the TP, but the back trajectories on 1 and 3 May illustrated that the airflows that ended at Lhasa were from the north or northwest rather than the south, indicating that the polluted dust over the TP on 3 May was more likely the mixing result of dust and smoke aerosol. In addition, the lengths of the back trajectories (especially the back trajectories at 10 m and 500 m above ground level) on 1 May showed that the airflows moved slowly, which allowed the possibility of aerosol mixture over the TP.*"

According to Jia et al. (2015), the dust from India transported to TP is mainly occurred in west region of TP and much less than that from Taklimakan Desert. In addition, the dust from India is generally polluted by anthropogenic aerosols (Jia et al., 2015). Theory, GEOS-Chem model can give the contributions of anthropogenic aerosol and dust aerosol through multi-group sensitivity experiments of controlling the related emission inventories in the research region. But, the results may be not reliable (especially for TP region) for the inventories and the chemical, mixing, aging, deposition processes and so on. And the evaluations of the model results need more measurement experiments and chemical observed data which are hard to obtain. This is not the goal of this manuscript. This question is worthwhile to study in the next step.

**Minor comments:**

1.Page 3, Line9-10, what is the meaning of "large scale"? Spatial scale or temporal scale? The sentence need be illustrated clearly.

Response: It has been corrected as "*large spatial scale*".

2.Page 3 Line 10, "satellite remote sensing method (Li et." should be "satellite remote-sensing method (Li et.", in which a space is needed between "method" and "(". In the whole manuscript, such writing problem should be paid attention, for example, Page 3, Line 20, there should be a space between "2007;" and "Xia", etc.

Response: All of them have been corrected.

3.Page 3, Line 27, there is mistake in grammar in sentence "there is an urgent need to. . .. . .".

Response: It has been corrected as "*it is very essential to...*".

4.Page 4, Line 3, "2.1 site" should be"2.1 Site".

Response: Corrected.

5.Page 4, Line 6, there is mistake in grammar in sentence "site where can suffer from the local anthropogenic emissions".

Response: It has been corrected as "*site that suffers from the local anthropogenic emissions*".

6.Before the unit, there need a space,for example, Page 4, Line 34, "2330km".

Response: All of these have been corrected in revised version.

7.Page 5 Line 41 and 42, 'Mt_WLG sites' should be 'Mt_WLG site'.

Response: It has been corrected.

**Response to Comments of Reviewer #2**

**General comments:** Tibetan Plateau (TP) plays a very important role in East Asian climate. Perturbation in thermodynamic fields of the Qinghai-Xizang Plateau by anthropogenic or natural aerosols might induce substantial regional climate changes and serious air pollutions. However, the variations of aerosols in TP region are less known compared with those in East or South Asian regions. This study investigates the characteristics and potential sources of aerosols in TP based on ground-based and satellite observations as well as numerical models. The results are interesting and they may help us better understanding the temporal and spatial variations of the aerosols in TP and subsequently the aerosol climate effects in Asian region. The topic of this study is novel to some degrees. And the paper has a potential for publication in the journal after revisions.

Response: Thanks a lot for your important comments and suggestions. We have made our best efforts to modify the manuscript according to your comments and suggestions.

**Comments:**

1. Introduction should be re-organized to a degree to make it more readable and more clearly.

Response: We have tried our best to re-organized the introduction and added some statements to make it more clearly, including as followed:

Moved the last four lines of first paragraph to the beginning of fourth paragraph in the revised version.

Separated the shortage of current study and the subject of this work (the third paragraph of the origin version), and added some sentences to show the research background in the third paragraph.

Added a connection sentence before the citation of Lau et al. (2006), i.e., "*The increase in aerosols over the TP may have an important impact on the regional or global climate.*"

Moreover, this paper has been edited by native English speakers to make it more readable.

2. The authors should make some comparisons of aerosol optical properties which derive from different platforms when investigating the temporal and spatial variations of aerosols in TP region.

Response: Thanks for this suggestion. We have added the comparison of aerosol optical properties between MODIS and CE318 sunphotometer in revised version at section 3.2 as following:

"

*Ground-based observations can offer accurate aerosol optical properties at point locations but lack spatial coverage. The MODIS aerosol product can provide the spatial variation in AOD over the TP. Thus, we evaluated the MODIS_AOD using the ground-based observation CE318_AOD at 550 nm over the TP sites. The CE318_AOD at 550 nm was interpolated from 440 nm, 675 nm, 870 nm and 1020 nm by using an established fitting method from Ångström (1929). The matchup method was that the CE318 data*

*within 1 hour of the MODIS overpass were compared with the MODIS data within a 25 km radius of the ground-based site. The minimum requirement for a matchup was at least 3 pixels from MODIS.*

*Figure 5 shows the results of MODIS_AOD compared to the collocated ground CE318 observations over the TP. There are 996 instantaneous matchups of Terra and Aqua MODIS during the CE318 instrument measurement period at the five TP sites. The MODIS_AOD overestimates the AOD at 550 nm with a positive mean bias of 0.02 and a root mean squared error (RMSE) of 0.11. The RMSE value is lower than that of the North China Plain sites (~0.25) (Bilal et al., 2019). The slope and intercept of the best-fit equation between the MODIS_AOD and CE318_AOD at 550 nm are 0.46 and 0.06, respectively, with a correlation coefficient (R) of 0.54. There are 67.8% of the compared AODs within the expected error envelope of 0.05+0.15AOD (%EE). The R value is lower than that in the global assessment statistics, while the %EE is higher than that in the global evaluation (Bilal and Qiu, 2018). Overall, the results suggest that the MODIS_AOD product can be used to study the aerosol spatial variation over the TP region.*

[Figure]

*Figure 5. Comparisons of the 550 nm AOD measured by the CE318 instrument (CE318_AOD) over Tibetan Plateau stations with the MODIS retrieval Deep-Blue/Dark-Target combined AOD of 10 km spatial resolutions (MODIS_AOD). The statistical parameters in this figure include the number of matchup data (N), the slope and intercept at the y-axis of linear regression (read line), the mean bias (MB), root mean squared error (RMSE), correlation coefficient (R), and the percentage of data within the expected error 0.05+0.15AOD (%EE) which is used as the MODIS AOD expected uncertainty over land (green lines).*"

3. A more detailed description on the accuracy of each type of platform data is needed.

Does MODIS products accurate enough in bright surface (such as in desert region in TP)?

Response: The accuracy of the data from ground-based CE318 instruments was shown in section 2.2.1, and we have added the calibration and data control in section 2.2.1, i.e., "*The instruments were periodically calibrated using the Langley method at AERONET global calibration sites (the Izaña, Spain or the Mauna Loa, USA) or using the inter-comparison calibration method at the Beijing-CAMS site (Che et al., 2015). The cloud-screened and quality-controlled data of AOD, ...*"

For CALIOP data, the data version is specified ("*version 4.10*") and a reference of data assessment has been cited in section 2.2.3, i.e., "*Kumar et al. (2018) have showed that the AOD from CALIOP version 4.10 agreed with the ground-based CE318 observation at a site in the central Himalayas with a correlation > 0.9 and ~ 87 % matchup data were within the expected error.*"

For the MODIS data, we used the MODIS Collection 6 Deep-Blue (DB)/ and Dark-Target (DT) combined AOD at 550 nm product. The description of this product has been added as followed: "*The MODIS AOD at 550 nm (MODIS_AOD) combined the DT and DB algorithms merges the products from the two algorithms based on the normalized difference vegetation index (NDVI) statistics as follows: 1) the DT AOD data are used for NDVI > 0.3; 2) the DB AOD data are used for NDVI < 0.2; and 3) the mean of both the algorithms or AOD data with high quality flag are used for 0.2 ≤ NDVI ≤ 0.3.*" Thus, the MODIS DT-DB AOD used the value from DB algorithm in bright surface, which algorithm is regarded as the better retrieval of AOD in bright surface than DT algorithm. In addition, we have added the evaluation of MODIS products using the ground CE318 sunphotometer observations, and the results showed that 67.8% of the compared AODs were within the expected error envelope of 0.05+0.15AOD. The content that added in section 3.2 can be seen in the response of comment 2.

4. Validation of GEOS-Chem is need. The authors should compare the simulated aerosols with the observations.

Response: The simple comparison between model simulated AOD and ground observed AOD was shown in figure 13. We wanted to validate the GEOS-Chem using MODIS AOD, but MODIS AOD products were almost unavailable over The TP for the cloud contamination during the case period. We have not data of observed chemical component, so this evaluation can not be conducted. But according this suggestion, we have added more evaluated parameters between model and CE318 observed data in the third paragraph of section 5 and the figure is updated in the revised version as followed: "*The evaluation results showed that the model underestimated the daily AOD at the three sites during this period, with negative mean biases from -0.28 to -0.08. However, the Model_AOD was relatively high correlated with the CE318_AOD at 550 nm, with the R values of 0.61 at Lhasa, 0.89 at NAM_CO and 0.86 at QOMS_CAS. These R values are higher than the model evaluation in South China and Indo-China Plain (~0.5)*

*(Zhu et al., 2017).*"

[Figure]

*Figure 1. The GEOS-Chem model simulated the daily average AOD vs CE318 observed*
*daily AOD at 550nm and the ratios of dust or organic carbon (OC) and black carbon*
*(BC) aerosol to the total AOD during 27 April, 2016 – 3 May, 2016 at Lhasa, NAM_CO*
*and QOMS_CAS. The statistical parameters used in Modal evaluation are the same as*
*Figure 5.*

5. How frequency of aerosol pollutions in Qinghai-Tibet Plateau based on your study?
Response: The frequencies of high aerosol loading (AOD 440 nm > 0.4) during the
CE318 observation period were 1.57%, 1.79%, 0.21%, 0.42% and 0.11% at the Lhasa,
Mt_WLG, Muztagh_Ata, NAM_CO, and QOMS_CAS site, respectively. These values
are relatively low. But as one of the most pristine terrestrial regions of the Earth, the
high aerosol loading over TP needs to be studied. The frequencies have been added in
the revised version, i.e., "*The frequencies of high aerosol loading (CE318_AOD > 0.4)*
*during the CE318 measurements were 1.57%, 1.79%, 0.21%, 0.42% and 0.11% at the*
*Lhasa, Mt_WLG, Muztagh_Ata, NAM_CO, and QOMS_CAS sites, respectively.*"

6. A deeper discussion is needed in Results section, such as make some comparisons or
summaries from similar studies.
Response: We have added some discussion by comparing to other studies, including but
not limited to:
The comparison of AOD in Tibetan sites and other regional background sites in China
is added in section 3.1.
"*The annual averages of CE318_AOD (shown in Figure 2) are 0.05-0.14 over TP sites.*
*These average values are lower than those in other regional background sites, such as*
*Longfengshan (0.35) in Northeast China (Wang et al., 2010), Xinglong (0.28) in North*
*China Plain (Zhu et al., 2014), Lin'an (0.89) in Eastern China (Pan et al., 2010) and*
*Dinghushan (0.91) in Southern China (Chen et al., 2014). The low aerosol loading over*
*the five TP sites indicates excellent air quality over the TP region.*"

The EAE in TP sites are compared with the inland urban and suburban sites in China
by adding the values of EAE.
"*This size distribution explained the relative low annual averages of EAE at the five*

*sites (all annual EAE in Figure 2 are less than <1.0), compared to the those at the inland urban and suburban sites in China (Xin et al., 2007), such as Beijing (1.19) (Fan et al., 2006), Nanjing (1.20) (Zhuang et al., 2018; Zhuang et al., 2017), Kunming (1.25) (Zhu et al., 2016), and Chengdu (1.09) (Che et al., 2015)*".

The results of the evaluation of MODIS AOD over TP are compared with the global and the other regional evaluations. See the response of comment 2.

The case study has been compared with another case study. The discussion of the difference from Jia et al. (2015) is added, i.e., "*Jia et al. (2015) has shown that the dust from India polluted by anthropogenic aerosols can be transported to the TP, but the back trajectories on 1 and 3 May illustrated that the airflows that ended at Lhasa were from the north or northwest rather than the south, indicating that the polluted dust over the TP on 3 May was more likely the mixing result of dust and smoke aerosol. In addition, the lengths of the back trajectories (especially the back trajectories at 10 m and 500 m above ground level) on 1 May showed that the airflows moved slowly, which allowed the possibility of aerosol mixture over the TP.*"

7. Conclusions should be shortened and more concise.
Response: The major conclusions have been refined as:
   "
(1) *The annual CE318_AOD at most TP sites showed increasing trends (0−0.013/year) during the past decade. Increasing tendencies in the annual-averaged EAE were also found at most TP sites. Spatially, the MODIS_AOD showed negative trends in the northwest edge close to the Taklimakan Desert and the east of Qaidam Basin and slightly positive trends in most of the other areas of the TP.*
(2) *Different aerosol types and sources contributed to the high aerosol loading at the five sites: dust was dominant in Lhasa, Mt_WLG and Muztagh with sources from the Taklimakan Desert, but fine aerosol pollution was dominant at NAM_CO and QOMS_CAS with the transport from South Asia.*
(3) *A case of smoke followed by dust pollution at Lhasa, NAM_CO and QOMS_CAS during 28 April – 3 May 2016 showed that the smoke aerosol in South Asia was first uplifted to 10 km and transported to the centre of TP. Then, the dust from the Taklimakan Desert could climb the northern slope of the TP and be transported to the TP, allowing the dust and smoke aerosol over the TP to mix.*
   "

8. English should be improved substantially throughout the whole manuscript.
  Response: The revised paper has been improved by native English speakers.

**Marked-up Manuscript:**

[revised manuscript text omitted]

---

## Author Response (AR2)

**Spatiotemporal variation of aerosol and potential long-range transport impact over Tibetan Plateau, China**

**Jun Zhu et al.**

We appreciate the reviewer's comments and suggestions. The manuscript has been revised accordingly. Our point-by-point responses to the comments are presented below. The comments are in black, followed by responses in blue and revised manuscript in *blue* with changes marked by underline.

**Response to Comments**

Comments to the Author:

The following studies are concerned with the aerosol transport over Tibetan Plateau that may be referred to in the introduction:

1. Page 15, Line 4-5:

(1) Liu, Y., et al., Aerosol optical properties and radiative effect determined from sky-radiometer over Loess Plateau of Northwest China, Atmos. Chem. Phys., 11, 11455–11463, 2011.

(2) Liu Y., et al., A review of aerosol optical properties and radiative effects. J. Meteor. Res., 28(6), 1003-1028, 2014.

(3) Huang J., et al., Possible influences of Asian dust aerosols on cloud properties and radiative forcing observed from MODIS and CERES. Geophysical Research Letters, 33, L06824, doi: 10.1029/2005GL024724, 2006a.

(4) Huang J., et al., Satellite-based assessment of possible dust aerosols semi-direct effect on cloud water path over East Asia. Geophysical Research Letters, 33, doi: 10.1029/2006GL026561, 2006b.

Response: We thanks for the comments about this manuscript. These studies have been cited.

2. Page 15, Line 7-8:

(1) IPCC (2013) Climate change 2013: The physical science basis. Stocker, T.F., et al. Eds., Contribution of Working Group I to the Fifth Assessment Report of the Intergovernmental Panel on Climate Change, Cambridge University Press, Cambridge, United Kingdom and New York, 1535 p.

Response: We have referred to this report.

3. Page 15, Line 18-21:

(1) Jia R., et al., Anthropogenic Aerosol Pollution over the Eastern Slope of the Tibetan Plateau. Advances in Atmospheric Sciences, 2019, 36(8): 847-862.

(2) Jia R., et al., Source and transportation of summer dust over the Tibetan Plateau. Atmospheric Environment, 123(2015), 210–219, doi:10.1016/j.atmosenv.2015.10.038, 2015.

(3) Zhu Q., et al., A numerical simulation study on the impact of smoke aerosols from Russian forest fires on the air pollution over Asia. Atmospheric Environment, 182, 263-

274, 2018

Response: These studies have been added after the sentence.

4. Page 15, Line 25:
Jia, R., et al., Estimation of the aerosol radiative effect over the Tibetan Plateau based on the latest CALIPSO product. J. Meteor. Res., 32(5), 707–722. doi: 10.1007/s13351-018-8060-3, 2018.

Besides the impact of aerosols over the TP on the radiation budeget, temperature and Indian summer monsoon, Liu et al., (2019a) reported a potential relationship may exist between the aerosol index and ice cloud properties over the TP, in which the aerosols have a more dominant influence than meteorological conditions on ice cloud properties (except for the nocturnal ice cloud droplet radius and ice water path during the daytime). (Liu, Y., et al., Effect of aerosols on the ice cloud properties over the Tibetan Plateau. Journal of Geophysical Research: Atmospheres, 124, 9594–9608, https://doi.org/10.1029/2019JD030463, 2019a)

Furthermore, Liu et al. (2019b) found the effect of the dust aerosols on the development of convective clouds and then on the precipitation over the downstream regions. Liu et al. (2019b) found that, with the AOD increasing to its peak in a dusty case over the TP, the ice particle size decreases to a minimum, convective clouds develop at higher heights because of the prolonged cloud life, and eastward movement of some polluted convective clouds could induce significant precipitation over the Yangtze River basin and North China. (Liu Y., et al., Impact of dust-polluted convective clouds over the Tibetan Plateau on downstream precipitation. Atmospheric Environment, 209, 67-77, 2019b).

Response: According to this comment, the effect of TP aerosol on cloud and precipitation in these studies has been cited and added as followed:

[revised manuscript text omitted]